# Reprogramming of cardiac phosphoproteome, proteome, and transcriptome confers resilience to chronic adenylyl cyclase-driven stress

Jia-Hua Qu[1,2†], Khalid Chakir[1], Kirill V Tarasov[1], Daniel R Riordon[1], Maria Grazia Perino[1], Allwin Jennifa Silvester[1], Edward G Lakatta[1]*

[1]Laboratory of Cardiovascular Science, Intramural Research Program, National Institute on Aging, National Institutes of Health, Baltimore, United States; [2]Department of Immunology, St. Jude Children's Research Hospital, Memphis, United States

*For correspondence:
lakattae@grc.nia.nih.gov

Present address: †Department of Host-Microbe Interactions, St. Jude Children's Research Hospital, Memphis, United States

Competing interest: The authors declare that no competing interests exist.

**Abstract** Our prior study (Tarasov et al., 2022) discovered that numerous adaptive mechanisms emerge in response to cardiac-specific overexpression of adenylyl cyclase type 8 (TGAC8) which included overexpression of a large number of proteins. Here, we conducted an unbiased phosphoproteomics analysis in order to determine the role of altered protein phosphorylation in the adaptive heart performance and protection profile of adult TGAC8 left ventricle (LV) at 3–4 months of age, and integrated the phosphoproteome with transcriptome and proteome. Based on differentially regulated phosphoproteins by genotype, numerous stress-response pathways within reprogrammed TGAC8 LV, including PKA, PI3K, and AMPK signaling pathways, predicted upstream regulators (e.g. PDPK1, PAK1, and PTK2B), and downstream functions (e.g. cell viability, protein quality control), and metabolism were enriched. In addition to PKA, numerous other kinases and phosphatases were hyper-phosphorylated in TGAC8 vs. WT. Hyper-phosphorylated transcriptional factors in TGAC8 were associated with increased mRNA transcription, immune responses, and metabolic pathways. Combination of the phosphoproteome with its proteome and with the previously published TGAC8 transcriptome enabled the elucidation of cardiac performance and adaptive protection profiles coordinately regulated at post-translational modification (PTM) (phosphorylation), translational, and transcriptional levels. Many stress-response signaling pathways, i.e., PI3K/AKT, ERK/MAPK, and ubiquitin labeling, were consistently enriched and activated in the TGAC8 LV at transcriptional, translational, and PTM levels. Thus, reprogramming of the cardiac phosphoproteome, proteome, and transcriptome confers resilience to chronic adenylyl cyclase-driven stress. We identified numerous pathways/function predictions via gene sets, phosphopeptides, and phosphoproteins, which may point to potential novel therapeutic targets to enhance heart adaptivity, maintaining heart performance while avoiding cardiac dysfunction.

## eLife assessment

This study describes **important** results from cardiac-specific overexpression of adenylyl cyclase type 8 (TGAC8) mice that was integrated with transcriptomic and proteomic evidence. The paper **convincingly** provides new insights into how one can interpret signals from visceral organs.

## Introduction

TGAC8 markedly increases the heart rate and ejection fraction, similar to acute exercise but chronic and in the absence of corporal exertion, which persists on a 24 hr per day and 7 days per week basis (*Lipskaia et al., 2000*; *Georget et al., 2002*; *Esposito et al., 2008*; *Moen et al., 2019*). Despite this constant high-level of heart performance driven by markedly elevated level of AC8-cAMP-PKA-Ca$^{2+}$ signaling, which typically proceeds to near-term cardiac dysfunction, the TGAC8 survives for an extended period of time, up to about one year of age without excessive mortality (*Tarasov, 2022*). Adaptive mechanisms that emerge in the TGAC8 heart confer resilience to this marked chronic stress of increased AC8-cAMP-PKA-Ca$^{2+}$ signaling and include the enrichment and activation of a large number of proteins involved in multiple signaling pathways related to nutrient sensing, energy production, growth factor signaling, and cell proliferation (*Tarasov, 2022*).

As the body is a complex system, multiple levels of coordinated regulatory mechanisms are requisite to defend against stimuli and maintain health (*Qu et al., 2022*; *Zhang et al., 2019*; *Pei et al., 2019*; *Deng et al., 2022*). Phosphorylation is one of the most important and has been the most extensively studied PTM of proteins. Various kinases and phosphatases within cells phosphorylate and dephosphorylate, respectively, a myriad of proteins to regulate their activities and affect multiple signaling pathways. Because the PKA signaling pathway was highly enriched in TGAC8 vs. WT (*Tarasov, 2022*), we reasoned that protein phosphorylation differed between TGAC8 and WT, and protein phosphorylation modification plays a crucial role in the remarkable adaptative capacity of the TGAC8 heart in response to markedly increased AC8-cAMP-PKA-Ca$^{2+}$ signaling.

By the utilization of TMT-labeling mass spectrometry (MS), we identified 12963 phosphopeptides and 3646 phosphoproteins, and observed a general upregulation of phosphorylation. By comparing genotypic differences in the proteome and phosphoproteome of the present study with genotypic differences in the transcriptome published previously (*Tarasov, 2022*), we sought to determine whether some adaptive mechanisms that emerge in the TGAC8 LV are regulated at transcriptional, translational, and PTM (phosphorylation) levels. We observed enrichments of phosphorylation, upregulation of kinase expression, and predicted activation of upstream kinases to expand our understanding of the regulatory mechanisms the TGAC8 utilizes to adapt to the chronic stress of activated AC8-cAMP-PKA-Ca$^{2+}$ signaling. We integrated the genotypic difference between the TGAC8 and WT of the phosphoproteome with the transcriptome and proteome, which broadened our understanding of how changes in the transcription, translation, and PTM of proteins in the TGAC8 LV may singly or coordinately enable the TGAC8 to endure the chronic AC8-induced stress.

## Materials and methods

### Animals

The TGAC8 mouse, generated by inserting a cDNA coding for human AC8 after the murine α-myosin heavy chain promoter, was a gift from Nicole Defer/Jacques Hanoune, Unite de Recherches, INSERM U-99, Hôpital Henri Mondor, F-94010 Créteil, France. The 3-month-old male TGAC mice were used as study cases, and age-matched WT littermates, in the C57/BL6 background, were used as controls. All studies were performed in accordance with the Guide for the Care and Use of Laboratory Animals published by the National Institutes of Health (NIH Publication no. 85–23, revised 1996). The experimental protocols were approved by the Animal Care and Use Committee of the National Institutes of Health (protocol #441-LCS-2016).

### Isolate and store mouse heart

Three-month-old male C57 mice (TGAC8 and WT) were euthanized, and the heart was isolated. The LV tissues were extracted and stored in tubes at –80 °C.

### Mass spectrometry for phosphoproteome and data process

TGAC8 and WT LVs were sent to the National Heart, Lung, and Blood Institute, National Institutes of Health, Bethesda to conduct MS using the standard tandem mass tag (TMT) labeling method to obtain the phosphoproteome and total proteome concurrently in the same platform. Mass spectrometry manipulation referred to our recently published paper (*Tarasov, 2022*).

Proteins from each group of samples are being quantified using the Bradford Assay Kit. Two-hundred micrograms of protein from each group will be labeled with Amine-Reactive Tandem Mass Tag Reagents (TMT Label Reagents; Thermo Scientific) according to the protocol supplied by the manufacturer. Briefly, protein will be denatured followed by reduction (TCEP) and alkylation (iodoacetamide), and then the alkylated proteins (200 ug) for each sample were precipitated in six volumes of ice-cold acetone at –20 °C overnight and later reconstituted in Triethylammonium bicarbonate. Each sample was digested overnight using Trypsin enzyme (enzyme to substrate ratio of 1:12.5) to get the peptide solution. Trypsin will cleave proteins at lysine and arginine amino acid residues yielding tryptic peptides.

The 10-plex TMT labeling was carried out according to Thermo Fisher Scientific's TMT Mass Tagging kits protocol. Each of the TMT 10-plex label reagents was reconstituted in Acetonitrile (ACN) and digested peptides from each sample were incubated with specific tags (126, 127 N, 127 C, 128 N, 128 C, 129 N, 129 C, 130 N, 130C, and 131, respectively) for about 1 hr at room temperature. The TMT labeled samples are then combined and fractionated using Off-Line high pH Basic Reverse Phase Fractionation to decrease the complexity of the sample and to increase the number of peptides being identified into a total of 24 fractions which will be later analyzed using LC-MS/MS. About 5% of the labeled tryptic peptides from each of the 24-fractions were used for global proteomics analysis by nanoLC-MS/MS analysis while the remaining 95% of samples in the 24-fractions were subjected to subsequent TiO2-enrichment for quantitative phosphoproteomics analysis.

TiO2 enrichment was conducted using a Titansphere Phos-kit (GL biosciences Inc). The TMT 10-plex tagged tryptic peptides were reconstituted in Buffer B (25% lactic acid in 1% TFA, 80% Acetonitrile). The TiO2 SpinTip (1 mg resin/10 ul tip) was incubated with the sample, washed using the wash buffer following the manufacturers protocol, and the phosphopeptides were eluted in elution buffer sequentially using 5% ammonium hydroxide solution and 5% pyrrolidine solution. The eluted fraction was dried and reconstituted in 0.1% FA for subsequent clean-up using the ZipTip C18-tips (Millipore). The sample from the elution fraction of C18-tips was dried and reconstituted in 0.1% FA for LC-MS/MS analysis.

The flow-through fraction and the wash fraction from the TiO2 spinTip were collected, pooled into a total of six fractions, and speedvac dried which will be later used for Fe-IMAC phosphoenrichment protocol. The peptide sample from six-pooled flow-through fractions from TiO2 enrichment underwent a second round of phosphoenrichment using a High-Select Fe-NTA Phosphopeptide Enrichment Kit (Thermo Scientific). The eluted fraction was dried and reconstituted in 0.1% FA for subsequent LC-MS/MS analysis.

Protein identification by LC-MS/MS analysis of peptides was performed using an Orbitrap Fusion Lumos Tribid mass spectrometer (Thermo Fisher Scientific, San Jose, CA) interfaced with an Ultimate 3000 Nano-HPLC apparatus (Thermo Fisher Scientific, San Jose, CA). Peptides were fractionated by EASY-Spray PepMAP RPLC C18 column (2 µm, 100 A, 75 µm × 50 cm) using a 120 min linear gradient of 5–35% ACN in 0.1% FA at a flow rate of 300 nl/min. The instrument was operated in data-dependent acquisition mode (DDA) using an FT mass analyzer for one survey MS scan on selecting precursor ions followed by top 3 s data-dependent HCD-MS/MS scans for precursor peptides with 2–7 charged ions above a threshold ion count of 10,000 with normalized collision energy of 37%. Survey scans of peptide precursors from 300 to 2000 m/z were performed at 120 k resolution and MS/MS scans were acquired at 50,000 resolution with a mass range m/z 100–2000.

All MS and MS/MS raw spectra of TMT experiments from each set were processed and searched using the Sequest HT algorithm within the Proteome Discoverer 2.2 (PD2.2 software, Thermo Fisher Scientific). The settings for precursor mass tolerance were set at 12 ppm, fragment ion mass tolerance to 0.05 Da, trypsin enzyme with 2 mis cleavages with carbamidomethylation of cysteine, TMT6-plex (lysine), TMT6-plex (peptide N-term) as fixed modification and deamidation of glutamine and asparagine, oxidation of methionine as variable modifications. The mouse sequence database from Swiss Sprot was used for the database search. Identified peptides were filtered for a maximum 1% FDR using the Percolator algorithm in PD 2.2 along with additional peptide confidence set to high. Only peptide spectra containing all reporter ions were designated as 'quantifiable spectra' and used for peptide/protein quantitation. The final lists of protein identification/quantitation were grouped and further filtered by PD 2.2 with at least two unique peptides per protein identified with high confidence. The quantitative protein ratios were weighted and

normalized by the median ratio for all quantifiable spectra of the peptides pertaining to the total protein identified.

For quantitative phosphopeptides analysis, an additional phosphorylation on Ser, Thr, and Tyr residues was specified as variable modifications. To confidently localize phosphorylation sites, the phosphoRS 3.0 node integrated in PD 2.2 workflow was used. The algorithm of phosphoRS 3.0 software enables automated and confident localization of phosphosites by determining individual probability values for each putatively phosphorylated site within the validated peptide sequences. For each relative ratio group of phosphopeptides/sites, no normalization was applied.

To remove low-quality signals, we remained peptides tagged with the medium and high confidence. We divided the expressions of phosphoproteins to those of total proteins to obtain the normalized phosphoproteome. As the expressions appeared to be near normal distribution, we calculated the fold change by dividing the average value of TGAC8 samples by the average value of WT samples, and calculated the p-value using two-sided unpaired Student's t-test. Data were deposited in MassIVE (ID: MSV000089850).

## Analyses of the TGAC8 LV transcriptome

The processed transcriptomic data were obtained from our recently published work (GEO: GSE205234) (*Tarasov, 2022*). The significantly changed mRNAs (TGAC8 vs. WT, q-value <0.05) were uploaded into DAVID (*Georget et al., 2002*) (version: 6.8; https://david.ncifcrf.gov/home.jsp) for gene ontology (GO) term enrichments, and were also imported into ingenuity pathway analysis (*Esposito et al., 2008*) (IPA (QIAGEN, March 2020)) for upstream kinases and phosphatases prediction. Kinases and phosphatases were selected from the transcriptome and their changes at mRNA level were displayed to observe their regulation by the overexpression of AC8.

## Analyses of phosphoproteome and proteome

Both unnormalized and normalized phosphoproteome were analyzed and visualized in PCA, heatmap, and volcano plots. With the cutoff -log(p-value)>1.3 and absolute(log2(fold change))≥0.4, we obtained the significantly changed unnormalized and normalized phosphoproteins (TGAC8 vs. WT) and imported them into IPA for functional analyses, respectively and compared the canonical pathway enrichments from these two datasets. In IPA, the upstream kinases were predicted based on the unnormalized phosphoproteom, and the regulatory pathways from the predicted upstream kinases to downstream effective signaling pathways were generated. Molecular and cellular functions were also enriched in IPA. The similar analyses to those conducted in the TGAC8 LV transcriptome were also done in the TGAC8 LV total proteome to observe the GO term enrichment, predicted activities of upstream kinases and phosphatases, and expression of kinases and phosphatases at the protein level.

## Analyses of phosphopeptides

The upregulated phosphopeptides (TGAC8 vs.WT, p-value <0.05 and log2(fold change)>0) were processed in R. After the split of peptides with multiple phosphosites into individual peptides and filtering of ambiguously identified sites, 2264 phosphopeptides were selected. Protein peptide sequences of full length were downloaded from the Uniprot database (*UniProt Consortium, 2019*) and matched to the protein accessions corresponding to processed and selected phosphopeptides. Peptides centered on the phosphosites and flanked by six amino acids on the N and C termini respectively were extracted from the matched protein peptide sequences, and uploaded into WebLogo (*Crooks et al., 2004*; http://weblogo.threeplusone.com/) and iceLogo (*Colaert et al., 2009*; https://iomics.ugent.be/icelogoserver/) for sequence motif analysis. In WebLogo online analysis, different amino acids were labeled in different colors representing different chemical properties (refer to figure legend). In iceLogo online analysis, a public amphiphilicity index of polar amino acids reported by Mitaku and colleagues (*Mitaku et al., 2002*) was used as an aid to characterize the amino acid preference at membrane-water interfaces. In addition, the phosphopeptides of PI3K/AKT substates were summarized and displayed in stacked bar plot.

## Integrative analyses of transcriptome, proteome, and phosphoproteome

The changes of significantly regulated molecules (TGAC8 vs. WT, q-value or p-value <0.05) in transcriptome, total proteome, and unnormalized phosphoproteome were displayed in the circos plot. In addition, the changes at different levels were compared and visualized in correlation scatter plots. The discordant proteins whose change directions in total proteome and phosphoproteome were opposite were used to analyze the specific regulation at the phosphorylation level by the use of Reactome and GO (Cellular Component and Biological Process) databases.

As to the coordinated regulation, the above mentioned significantly regulated molecules (TGAC8 vs. WT) in each dataset were imported into IPA for canonical pathway enrichment. PI3K/AKT, ERK/MAPK, and protein ubiquitination signaling pathways were coordinated across all omics. The phosphopeptides of protein substrates of PI3K were extracted from the phosphopeptide dataset and the numbers of changed phosphopeptides (TGAC8 vs. WT) were summarized and visualized in a stacked bar plot. The PI3K/AKT signaling pathway was visualized in IPA, and nodes in the pathways were marked in different colors in terms of their changes at different levels.

As kinases and phosphatases are pivotal in phosphorylation, and transcription factors (TFs) are important in transcription regulation, they were extracted from transcriptome, proteome, and phosphoproteome to observe their changes (TGAC8 vs. WT) and deduce their role in the reprogram of phosphoproteome. The upstream transcription regulators including but not only TFs were predicted on the basis of transcriptome and proteome by the utilization of a knowledge base in IPA, and their expressions were searched in multiple datasets. Furthermore, TFs that were significantly regulated at phosphorylation level (TGAC8 vs. WT, p-value <0.05) were defined as significantly changed phosphorylated TFs in this research, and they were uploaded into STRING database (*Moen et al., 2019*) (version: 11.0; http://string-db.org/) to generate protein-protein interaction network and enrich KEGG pathways. They were also imported into IPA for canonical pathway, molecular, and cellular functions enrichments.

## Western blot

Snap-frozen LV tissue from three-month-old male mice was homogenized and lysed in ice-cold RIPA buffer (Thermo Fisher Scientific: 25 mM Tris-HCl (pH 7.6), 150 mM NaCl, 1% NP-40, 1% sodium deoxycholate, 0.1% SDS) supplemented with a Halt protease inhibitor cocktail (Thermo Fisher Scientific), Halt phosphatase inhibitor cocktail (Thermo Fisher Scientific) and 1 mM phenylmethyl sulfonyl fluoride, using a Precellys homogenizer (Bertin Instruments) with tissue homogenization kit CKMix (Bertin Instruments) at 4 °C. Extracts were then centrifuged at 10,000 × g for 10 min at 4 °C and the protein concentration of the soluble fraction determined using the Bicinchoninic Acid (BCA) Assay (Thermo Fisher Scientific). Samples were denatured in Laemmli sample buffer (BioRad Laboratories) containing 355 mM 2-mercaptoethanol at 95 °C for 5 min, and proteins (10–50 µg/lane) resolved on 4–20% Criterion TGX Stain Free gels (Bio-Rad Laboratories) by SDS/PAGE. Gels were then exposed to UV transillumination for 2.5 min to induce crosslinking of Stain Free gel trihalo compound with protein tryptophan residues. Proteins were then transferred to low-fluorescence polyvinylidene difluoride (LF-PVDF) membranes (BioRad Laboratories) using an electrophoretic transfer cell (Mini Trans-Blot, Bio-Rad). Membrane total protein was visualized using an Amersham Imager 600 (AI600) (GE Healthcare Life Sciences) with UV transillumination to induce and a capture fluorescence signal.

Blocked membranes (5% milk/tris-buffered saline with Tween-20, TBST) were incubated with the following primary antibodies: (1) Anti-Phospho-PKA Substrate (RRXS*/T*) antibody, Cell Signaling Technologies, Cat.#9624, 1:1000 dilution; (2) Anti-Ubiquitin antibody produced in rabbit, Sigma-Aldrich, SAB1306582, 1:200 dilution; (3) Anti-pan-AKT antibody, Cell Signaling Technologies, Cat.#4691, 1:1000 dilution; (4) Anti-Phospho-pan-AKT (Thr308) antibody, Cell Signaling Technologies, Cat.#13038, 1:1000 dilution; (5) Anti-Phospho-pan-AKT (Ser473) antibody, Cell Signaling Technologies, Cat.#4060, 1:1000 dilution; (6) Anti-AKT2 antibody, Abcam, Cat.#13038, 1:1000 dilution; (7) Anti-Phospho-AKT2 (Ser473) antibody, Cell Signaling Technologies, Cat.#ab131168, 1:1000 dilution; (8) Anti-ERK1/2 antibody, Cell Signaling Technologies, Cat.#4695, 1:1000 dilution; (9) Anti-Phospho-ERK1/2 (Thr202/Tyr204) antibody, Cell Signaling Technologies, Cat.#4376, 1:1000 dilution; (10) Anti-p21 Ras antibody, Cell Signaling Technologies, Cat.#3965, 1:1000 dilution; (11) Anti-GAPDH antibody, Santa Cruz Biotechnology, Cat.# sc-32233, 1:1000 dilution; (12) Anti-c-Raf antibody, Thermo

Fisher Scientific, Cat.# PA5-29333, 1:1000 dilution; (13) Anti-Phospho-c-Raf (Ser338) antibody, Cell Signaling Technology, Cat.# 9427, 1:1000 dilution; (14) Anti-MEK1/2 antibody, Cell Signaling Technology, Cat.# 9126, 1:1000 dilution; (15) Anti-Phospho-MEK1/2 (Ser221) antibody, Cell Signaling Technology, Cat.# 2338, 1:1000 dilution.

Primary antibodies were then detected using horseradish peroxidase (HRP) conjugated antibody (Invitrogen) at 1:10,000. Bands were visualized using Pierce SuperSignal West Pico Plus ECL substrate kits (Thermo Scientific), the signal captured using an Amersham Imager 600 (AI600) (GE Healthcare Life Sciences) and quantified using ImageQuant TL software (GE Healthcare Life Sciences). Band density was normalized to total protein. Two-tailed unpaired Student's t-test with Welch's correction was used for statistics.

## Protein synthesis detection

Protein synthesis was assessed by SUnSET-Western Blot as previously described in the above section. Briefly, the puromycin solution was prepared in PBS, sterilized by filtration, and a volume of 200 µl was injected in mice intraperitoneally, to achieve a final concentration of 0.04 µmol/g of body mass. After 30 min, mice were sacrificed, the LV was harvested, and snap-frozen in liquid nitrogen. Protein extraction was performed using Precellys, quantified with BCA assay 25 µg of total protein were separated by SDS-PAGE; proteins were then transferred onto PVDF membrane and incubated overnight in the anti-puromycin primary antibody (MABE343, Sigma-Aldrich, St. Louis, MO). Visualization of puromycin-labeled bands was obtained using horseradish peroxidase conjugated anti-mouse IgG Fc secondary antibody (Jackson ImmunoResearch Laboratories Inc, West Grove, PA, USA), using Pierce Super Signal ECL substrate kit (Pierce/Thermo Scientific Rockford, IL). Chemiluminescence was captured with the Imager AI600 and densitometry analysis was performed using ImageQuantTL software (both by GE, Boston, MA). Total protein was used as a control for protein loading. Genotypic differences of protein synthesis were tested via an as unpaired t-test.

## Proteasome activity detection

Flash-frozen tissue was homogenized in ice-cold cytosolic extraction buffer (50 mM Tris-HCl pH 7.5, 250 mM Sucrose, 5 mM MgCl2, 0.5 mM EDTA, and 1 mM DTT). A bicinchoninic acid (BCA) assay (Pierce) was used to determine the protein concentrations. All samples were equally concentrated in proteasome assay buffer (50 mM Tris-HCl pH 7.5, 40 mM KCl, 5 mM MgCl2, and 1 mM DTT). Proteasome activity was determined in the presence of 28 µM ATP using the Suc-LLVY-AMC (18 µM, Boston Biochem #S280) fluorogenic substrate with and without proteasome inhibition (MG 132, 1 mM, Sigma). The plate was read at an excitation wavelength of 380 nm and an emission wavelength of 469 nm using a Spectramax M5 (Molecular Devices). Activity was calculated by subtracting the background (proteasome inhibited value) from the reading (proteasome activated value).

## Protein soluble and insoluble fractions detection

Protein aggregates were measured using Proteostat (Enzo, ENZ-51023) following the manufacturer's instructions. For this assay left ventricle myocardial lysate (Cell Signaling lysis buffer) was obtained, protein concentration was assayed (BCA assay [Pierce]). Ten µg of protein was loaded into a 96-well microplate and protein aggregates were analyzed using the Proteostat assay kit (Enzo Life Sciences) following the manufacturer's instructions. Background readings were subtracted from sample recordings and were normalized to wild-type values.

## Data process and graphics production

Most raw data were precleared and processed using RStudio (version: 1.1.463) in R language (version: 3.5.3), in which the tidyr package and dplyr package were applied. In addition, GraphPad Prism (version: 7), Microsoft Excel (version: 2019), and Adobe Illustrator (version: CC 2019) were also used for statistics and graphics. Venn diagrams were generated in Venny (*Tarasov, 2022*) (version: 2.1.0; http://bioinfogp.cnb.csic.es/tools/venny/index.html). Circos plot was generated using Circos software (*Qu et al., 2022*) in Perl language.

## Results

### The TGAC8 reprograms the LV phosphoproteome

The TMT-labeling quantitative MS identified **12963 phospho-peptides** (File 1: Sheet 1) and **3646 phosphoproteins** (File 1: Sheet 2). The principal component analysis (PCA) (*Figure 1A*) of the 3646 phosphoproteins in each mouse showed that the TG and WT samples were distinguished into two groups completely. The heatmap (*Figure 1B*) showed the intra-group consistency of upregulated phosphoproteins in TGAC8 compared to WT. Within the cutoff at -log10(p-value)>1.3 and absolute value of log2(fold-change(TG/WT))≥0.04, 781 phosphoproteins were differentially regulated by genotype. Of the 781 phosphoproteins, the phosphorylation levels of 741 proteins were increased and only 40 were decreased in TGAC8 vs. WT LV (*Figure 1C*).

### Functional analyses of the phosphoproteome

We performed the canonical pathway, upstream regulation, and downstream function analyses of the remodeled LV phosphoproteome of TGAC8 using the IPA software.

#### Canonical pathway analysis of the remodeled LV phosphoproteome of TGAC8

To study the mechanisms at phosphorylation level, we first performed canonical pathway analysis using IPA. Top canonical pathways within the phosphoproteome that were statistically differentially enriched in TGAC8 vs. WT were displayed in the bar plot (*Figure 1D* and File 2: Sheet 1). Of these pathways, cardiac hypertrophy signaling (enhanced) is most highly activated (z-score as high as 4.867) (*Figure 1D*). The combination of (*Figure 1—figure supplement 1*) and File 2: Sheet 2 and our recently published work (*Tarasov, 2022*) showed that the phosphorylation status of numerous molecules, e.g., Ca²⁺ ATPase, RyR2, Ncx, involved in excitation, calcium cycling, contraction, relaxation, and protein synthesis, was elevated in the TGAC8 vs. WT LV. However, we have demonstrated that at 3-month of age, the mouse age in the present study, the LV mass is not elevated in TGAC8 vs. WT (*Tarasov, 2022*). Of note, transcriptomic, proteomic, and WB analyses indicated that pathologic hypertrophic markers were not increased in TGAC8 vs. WT (*Tarasov, 2022*). In this regard, the activation of key stress response signaling pathways, e.g., protein kinase A (PKA) signaling, PI3K signaling, and AMPK signaling, may protect against pathological changes, that result in an increase in cardiac mass in TGAC8 despite its chronic increase in performance. For example, activated PKA signaling can restrict the activity of AC8, acting as a feedback to fine-tune the downstream signaling networks inside the body (*Willoughby et al., 2012*). Additionally, nitric oxide (NO) signaling may coordinate cardiac protection in the increase in cardiac mass (*Figure 1D*). Increased phosphorylation status of HSP90 and Akt in TGAC8 synergistically increase eNOS activity (*Takahashi and Mendelsohn, 2003*), and increased NO produced by eNOS may modulate the function of the ryanodine receptor Ca²⁺ release channel (RyR2) on the cardiac sarcoplasmic reticulum (*Lim et al., 2008*). The TMT-labeling MS identified significantly upregulated phosphorylation in the peptides of RyR2 (RISQTSQVSIDAAHGYSPR, E9Q401 [2805–2823]) (File 1: Sheet 1), consistent with the PKA-induced phosphorylation of RyR2-S2809 mediating the increase in Ca²⁺ spark amplitude (*Meissner, 2010*).

#### Upstream regulators of the remodeled LV phosphoproteome of TGAC8

As known, the overexpression of AC8 activates the AC8-cAMP-PKA phosphorylation axis (*Moen et al., 2019*). There were 2264 phosphosites whose names and locations are unambiguous within the 4990 upregulated phosphopeptides (p-value ≤0.05 and log2(fold-change(TG/WT))≥0) in TGAC8 vs. WT. The majority of these phosphosites were serines (S) and threonines (T), and a few were tyrosines (Y) (*Figure 2A*). By using WebLogo, we profiled the phosphorylation motif sequences (*Figure 2B*). A typical PKA and AKT substate motif, R-R-X-pS/T, was the most abundant substrate motif, consistent with increased PKA-dependent phosphorylation (*Figure 2B–D*; *Hunzicker-Dunn et al., 2012*; *Bellis et al., 2009*). In addition to S and T, the center phosphosites also included phosphotyrosine (Y). The center phosphosites were mainly flanked by the P-E-E sequence on the C terminus, endowing a strong hydrophobic and acidic property to the C terminus of the substrate sequences (*Figure 2B*), supported by the amphiphilicity index analysis in iceLogo (*Figure 2D*). Because phosphorylation substrate motifs were not restricted to PKA's substrate motif, additional kinases must be also involved

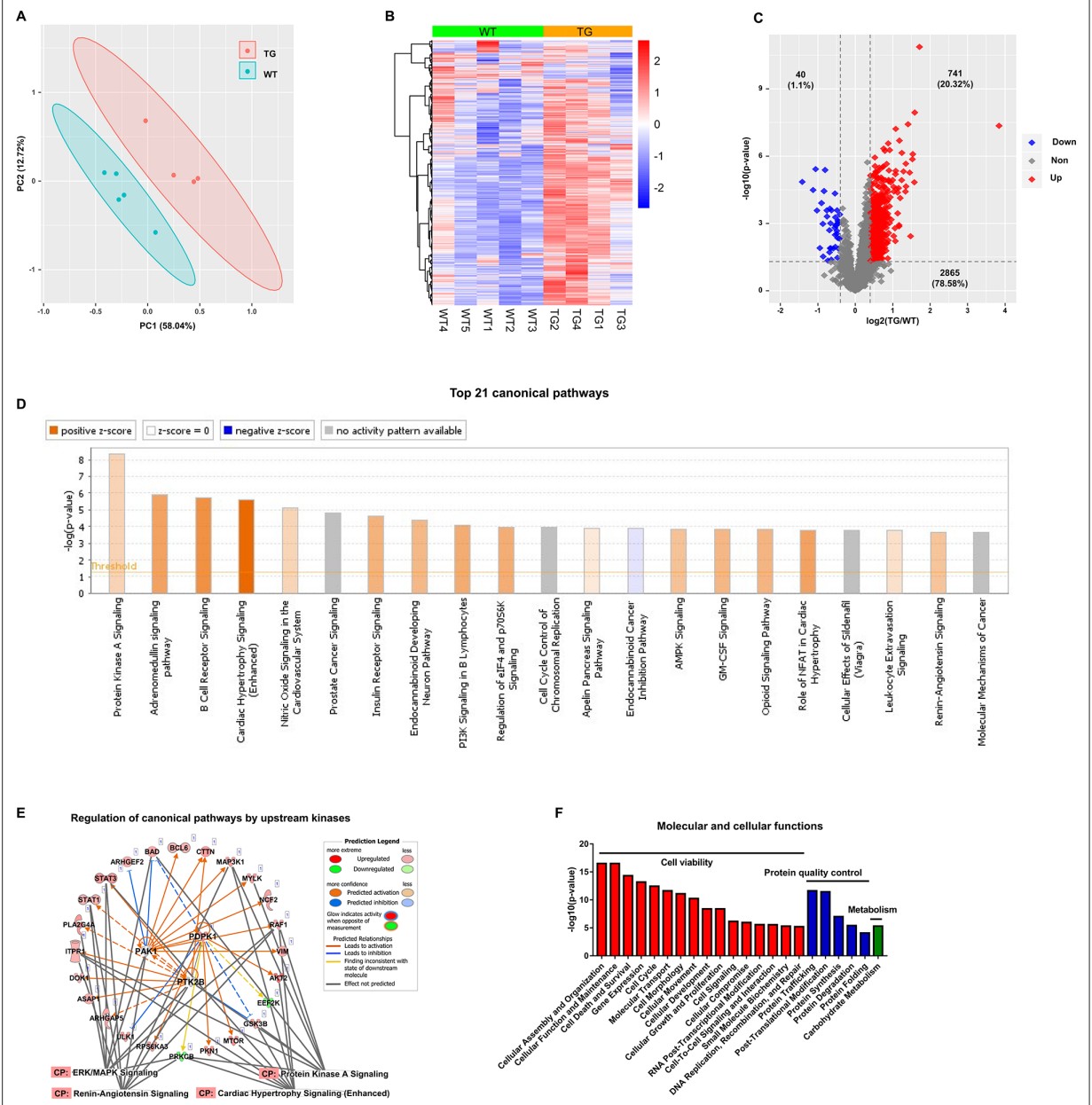

**Figure 1.** Cardiac-specific overexpression of adenylyl cyclase type 8 (TGAC8) reprograms the left ventricle (LV) phosphorylation state. (**A**) Principal component analysis (PCA) plot of phosphoproteome in TGAC8 vs. WT LVs. (**B**) Heatmap of scaled expressions of phosphorylated proteins. (**C**) Volcano plot of phosphoproteome. Red dots represent upregulated phosphoproteins with -log10(p-value)>1.3 and log2(fold change)≥0.4; blue dots represent downregulated phosphoproteins with -log10(p-value)>1.3 and log2(fold change)≤–0.4; gray dots represent nonsignificantly regulated phosphoproteins with -log10(p-value)≤1.3 or absolute(log2(fold change))<0.4. (**D**) Top 21 canonical pathways enriched from phosphoproteome. Orange and blue bars represent activated and inhibited pathways, respectively. The darker the color is, the greater the absolute(z-score) is, and the more significantly enriched the pathway is. (**E**) Regulatory pathway from upstream kinases. Three dots in the center represent the predicted upstream kinases. Surrounding dots represent identified phosphoproteins differentially regulated in phosphoproteome. Terms on the bottom represent canonical pathways in which the phosphoproteins are involved. Meanings of colors refer to the box of prediction legend besides the graph. (**F**) Enriched molecular and cellular functions from the phosphoproteome. They are related to three categories, including cell viability, protein quality control, and metabolism.

The online version of this article includes the following figure supplement(s) for figure 1:

**Figure supplement 1.** The enriched pathway, Cardiac Hypertrophy Signaling (Enhanced), from the phosphoproteome.

**Figure supplement 2.** The regulatory pathway is under the control of a predicted upstream kinase, PDPK1.

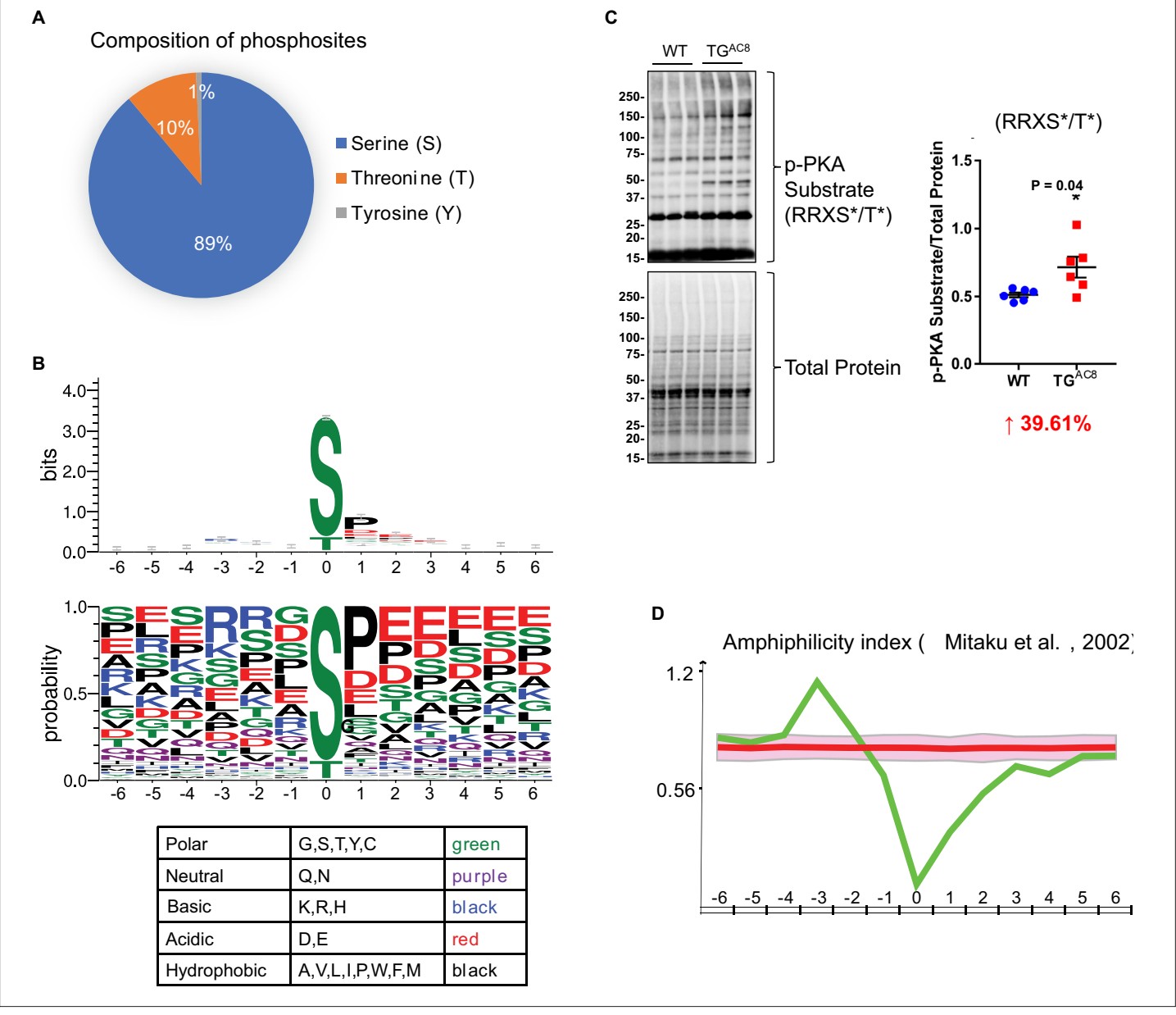

**Figure 2.** Characteristics of upregulated phosphopeptides. (**A**) Composition of phosphosites that reside on the upregulated phosphopeptides. (**B**) Sequence logo diagrams of upregulated phosphopeptides centered on the phosphosites and flanked by six amino acids on the N and C termini, respectively using WebLogo online software. Colors of amino acids represent different chemical properties. (**C**) Western blots of phosphorylation of protein kinase A (PKA) substrates and their statistics results: RRXS*/T*. Unpaired Student's t-test with Welch's correction was used.(**D**) Test the amphiphilicity feature of the upregulated phosphopeptides using iceLogo online software and refer to a published work by Mitaku and colleagues in 2002 (*Mitaku et al., 2002*).

in reprogramming the TGAC8 phosphoproteome. Therefore, we were inspired to explore additional kinase-substrate pathways mediating the effect of the cardiac-overexpression of AC8 here.

IPA predicted that 26 kinases were responsible for the above-mentioned phosphorylation motifs (File 2: Sheet 3), among which, three had previously been identified to differ by genotype in the LV transcriptome, 18 were shown to differ by genotype in the LV proteome, and 18 were identified in the present phosphoproteome. Of note, most of the identified kinases were upregulated in TGAC8 vs. WT at transcription, translation, and PTM (phosphorylation) levels. Within the phosphoproteome, eight kinases were predicted to be differentially activated by genotype, and the phosphorylation level of four of these (PDPK1, PAK1, PTK2B, and GSK3B) was increased in TGAC8 vs. WT. These four

kinases each phosphorylates various proteins to regulate cellular functions. For example, a previous study observed that in human glioblastoma cell lines, T98G and U87MG cells, the interference of human PDPK1 mRNA by siRNA decreased the phosphorylation level of human BAD protein (S112, S136, and S155) and activated cell death to suppress tumorigenesis (*Desai et al., 2011*). Another study reported that in mouse myoblast cell line, C2C12 cells, PDPK1 protein phosphorylated the mouse ULK1 protein (*Li et al., 2018*). Increased phosphorylation states of BAD (upregulated at S112, S136, and S155) and ULK1 (upregulated at S450, S521, and S637/638) in TGAC8 may be linked to activated PDPK1.

Different kinases can coordinate to reprogram the complex regulation network in the organism. Consistent with the IPA canonical pathway analysis, the phosphorylation level of 27 proteins (including the three kinases (PDPK1, PAK1, and PTK2B) auto-phosphorylating themselves) were elevated by the three kinases to regulate cardiac stress signaling pathways, including PKA signaling, ERK/MAPK signaling, cardiac hypertrophy signaling (enhanced), and renin-angiotensin signaling (*Figure 1E* and *Figure 1—figure supplement 2*), to cope with the cardiac stress in the TGAC8. For example, GSK3B is involved in multiple biological processes. PDPK1 phosphorylated GSK3B at S9 (*Chen et al., 2013*; *Ericson et al., 2010*) and PTK2B phosphorylated GSK3B at S9 and Y216 (*Hartigan et al., 2001*; *Tsai et al., 2009*) to inhibit the kinase activity of GSK3B. In this phosphoproteome, GSK3B was phosphorylated at multiple sites, including S9, Y216, S219, and S545, and thus GSK3B inhibition was linked to the cardiac adaptation (*Juhaszova et al., 2004*; *Juhaszova et al., 2009*).

## Downstream functions of the remodeled LV phosphoproteome of TGAC8

To study the downstream functions of these phosphoproteins, we performed molecular and cellular functions analysis in IPA. The enriched functions were mainly in three categories, i.e., cell viability, protein quality control, and metabolism (*Figure 1F* and File 2: Sheet 5). The functions of cellular assembly and organization, cellular function and maintenance, and cell death and survival were ranked as the top three terms, indicating the important role of the reprogrammed phosphoproteome in improving cell viability in response to the chronic stress of the overexpression of AC8 in the mouse heart.

## Integrative analyses of transcriptome, proteome, and phosphoproteome

We next integrated changes in PTM (phosphorylation) in TGAC8 vs. WT with genotypic differences in protein translation and in transcription. The circos plot (*Figure 3A*) illustrates the genotypic differences (TGAC8 vs. WT) in **molecules** identified across the transcriptome, proteome, and phosphoproteome. In addition, the correlation scatter plots (*Figure 3B*) demonstrated a moderate level of correlation of genotypic differences of molecular changes within phosphoproteome and transcriptome, and a somewhat stronger correlation between genotypic differences of molecular changes within phosphoproteome and proteome (*Figure 3C*). Concordant and discordant enrichments of canonical pathways differed by genotype among the three omics are shown in *Figure 3D*. A comparison of genotypic differences within the three omics highlights the importance of protein phosphorylation, and the phosphorylation was **coordinated with and supplemental to** transcription and translation.

Because kinases and phosphatases are the key regulators of phosphorylation, and phosphorylation of TFs are the primary determinant of their activities in transcription, we compared genotypic differences in kinases, phosphatases and TFs within the phosphoproteome and proteome with our previously published genotypic differences in the TGAC8 vs. WT transcriptome (*Tarasov, 2022*; *Figure 3E*). We identified 196 kinases, 71 phosphatases, and 191 TFs in the present phosphoproteome (*Figure 3F*). The protein phosphorylation status of about one-half of these molecules differed significantly (p-value <0.05) by genotype (*Figure 3G*), and the protein phosphorylation status of majority of these molecules was increased (*Figure 3H–J*). The genotypic differences in the transcription (mRNA expression obtained from RNA-seq) and translation (protein expression obtained from mass spectrometry) of these molecules, however, differed from those in the phosphoproteome (*Figure 3E*). Because there were more molecules that were unchanged or reduced in the TGAC8 vs. WT transcriptome or proteome than in the phosphoproteome, it is clear that the changes in protein phosphorylation are of utmost importance in the coordination of transcription, translation, and PTM, that underlie the adaptive response of the TGAC8 LV to the marked, chronic increase stress of its intrinsically activated

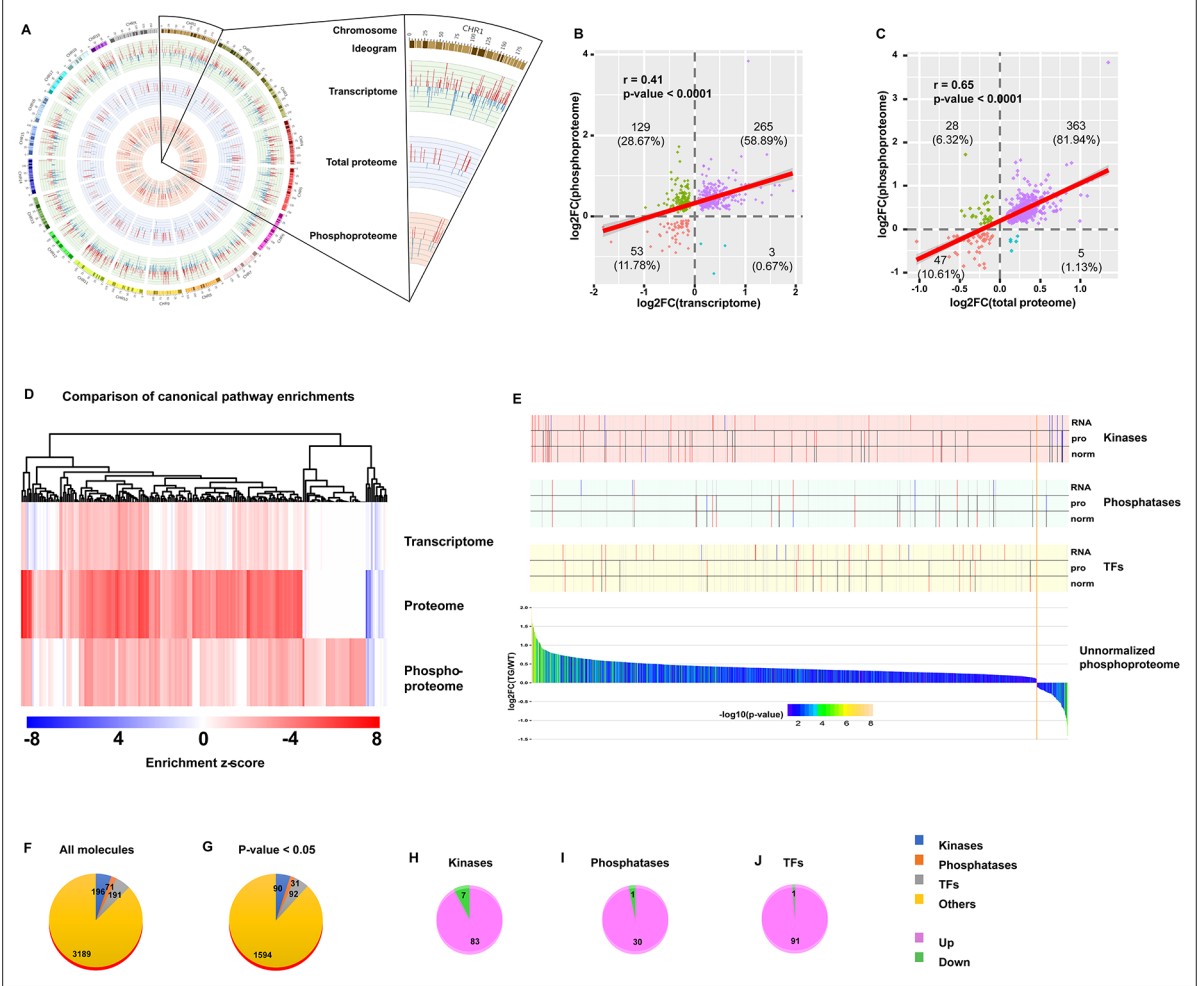

**Figure 3.** Integrative analysis of transcriptome, proteome, and phosphoproteome. (**A**) Circos plot of gene expression at different levels from transcription, to protein, and to phosphorylation. Bars in each layer of circle represent changes in significantly regulated molecules (q-value or p-value <0.05) in transcriptome, total proteome, and unnormalized phosphoproteome. Bar height, color, and direction represent log2(fold change). Red and upward indicate upregulation, and blue and downward indicate downregulation. (**B–C**) Scatter plots of correlation between phosphoproteome and transcriptome (**B**) or total proteome (**C**). (**D**) Comparison of canonical pathway enrichments from the transcriptome, total proteome, and phosphoproteome. Enrichment z-scores were calculated in IPA, and were scaled to generate heatmap in R language. (**E**) Integrative graph of expression of all phosphoproteins and selected types of molecules. Bar plot on the bottom displays expression of all significantly changed (p-value <0.05) phosphoproteins in unnormalized phosphoproteome. Y-axis denotes log2(fold change), and x-axis denotes each molecule ranked by log2(fold change). Bars were colored by -log10(p-value). Stick plots on the top display kinases, phosphatases, and transcription factors (TFs) selected from the bottom bar plot and their changes in transcriptome (RNA), total proteome (pro), and normalized phosphoproteome (norm). From top to bottom: each large block denotes the changes of certain types of molecules (from kinases to phosphatases, to TFs). In each block, the three rows from top to bottom indicate the changes in transcriptome, total proteome, and normalized phosphoproteome. Each stick represents a molecule and color represents the change: red: (p-value or q-value <0.05) and log2FC >0; blue: (p-value or q-value <0.05) and log2FC <0; black: (p-value or q-value ≥0.05) or log2FC = 0; gray: unidentified in one of the omics. Their locations are matched to those in the bottom bar plot. (**F–J**) Pie plots of the composition of phosphoproteins in the unnormalized phosphoproteome. Regardless of p-value, numbers of four types of proteins, including kinases, phosphatases, TFs, and others, are displayed in (**F**). After filtration with cutoff p-value <0.05, numbers of those types of proteins are displayed in (**G**). Extracted from G, numbers of upregulated and downregulated kinases (**H**), phosphatases (**I**), and TFs (**J**) are displayed.

AC-cAMP-PKA-Ca$^{2+}$ signaling. Thus, by utilizing multiple bioinformatic analysis methods, we not only dissected the reprogrammed phosphoproteome and deduced its functions in the TGAC8 mouse LV, but also profiled the coordination of phosphoproteome, transcriptome, and proteome.

## Implications of coordinated changes in protein phosphorylation and transcription in TGAC8 vs. WT

The most likely type of coordination between protein phosphorylation and transcription is related to phosphorylation states of TFs which is determined by kinases and phosphatases activities. Analysis of the remodeled LV transcriptome of TGAC8 in GO analysis using DAVID online software discovered that enrichments of numerous phosphorylation and dephosphorylation-related biological processes and molecular functions significantly differed by genotype (*Figure 4A*), and that the coordinated expression of numerous kinases and phosphatases also differed by genotype (*Figure 4B–C* and File 8). Because expression levels of kinases and phosphatases do not denote their activation states, we utilized IPA to predict the genotypic differences in activation states of upstream kinase and phosphatase (regulators of the transcriptome). Intriguingly, 60 kinases were predicted to be upstream regulators of the remodeled LV transcriptome of TGAC8, and the majority of these kinases were predicted to be activated in TGAC8 vs. WT (*Figure 4D* and File 9). In contrast, much fewer phosphatases were predicted between TGAC8 and WT, and most of these were predicted to be inhibited (*Figure 4E* and File 9). Thus, the differential genotypic enrichment of phosphorylation status and predicted activation of kinases and phosphatases identified in our post hoc analysis of our previously reported TGAC8 vs. WT transcriptome (*Tarasov, 2022*) are linked to the differential genotypic reprogramming of the phosphoproteome discovered in the present study.

We utilized IPA to predict the activation states of upstream transcription regulators on the basis of the remodeled LV transcriptomes and proteomes, and then determined their expression levels in transcriptome, proteome, and phosphoproteome (*Figure 4F*). Because TFs are the main regulators of transcription, and because phosphorylation of TFs usually regulates their activities (*Bohmann, 1990*; *Weigel and Moore, 2007*; *Mayr and Montminy, 2001*), we focused our subsequent analysis on functions of those TFs that were differentially phosphorylated in TGAC8 vs. WT. To confirm the hypothesis that the TGAC8 might exert transcriptional control by TFs phosphorylation, we retrieved all TFs identified in our previous transcriptome and proteome (*Tarasov, 2022*) and mapped them to the current phosphoproteome in order to determine their genotypic differences in phosphorylation status. Of the 191 TFs also identified in this phosphoproteome, 92 TFs differed significantly by genotype in phosphorylation status (p-value <0.05). In agreement with the general upregulated transcription in TGAC8 LV (*Tarasov, 2022*), nearly all of the TFs that differed by genotype had increased phosphorylation levels (*Figure 4G* and File 3). The only exception was purine-rich element binding protein G (Purg), known to be hyper-phosphorylated in solid tumors (*Zanivan et al., 2008*).

We conducted Kyoto Encyclopedia of Genes and Genomes (KEGG) pathway analysis to further probe the predicted facts of TFs that were generally hyper-phosphorylated in TGAC8 vs. WT. Numerous immunity-related pathways, cancer-related pathways, and metabolism-related pathways were enriched (*Figure 4H* and File 4). We then utilized IPA for in-depth analysis. Numerous metabolic canonical pathways (glucocorticoid receptor signaling and PPAR signaling), a cancer-related canonical pathway (regulation of epithelial-mesenchymal transition pathway), and immunity-related canonical pathways (role of JAK1, JAK2, and TYK2 in interferon signaling, IL-9 signaling, JAK/Stat signaling, etc.) were enriched (*Figure 4I* and File 5: Sheet 1). Intriguingly, the sumoylation pathway was markedly inhibited. Because both sumoylation and phosphorylation are important PTMs in transcriptional regulation (*Tomanov et al., 2018*; *Khan et al., 2014*), it is possible that the upregulated protein phosphorylation and inhibited protein sumoylation pathway in TFs coordinately activate transcription and improve immunity in the TGAC8 LV. In addition, cell viability and metabolism in molecular and cellular functions were enriched in TGAC8 vs. WT (*Figure 4J* and File 5: Sheet 2), consistent with enrichments of aforementioned KEGG pathways and IPA canonical pathways (*Figure 4H and I*).

Taken together, we have discovered the existence of substantial crosstalk between protein phosphorylation and transcription in the TGAC8 LV: phosphorylation of TFs reprograms immunity and metabolism-related pathways and functions in the TGAC8 LV, likely serving to maintain cell survival in response to the chronic stress of marked AC8 overexpression.

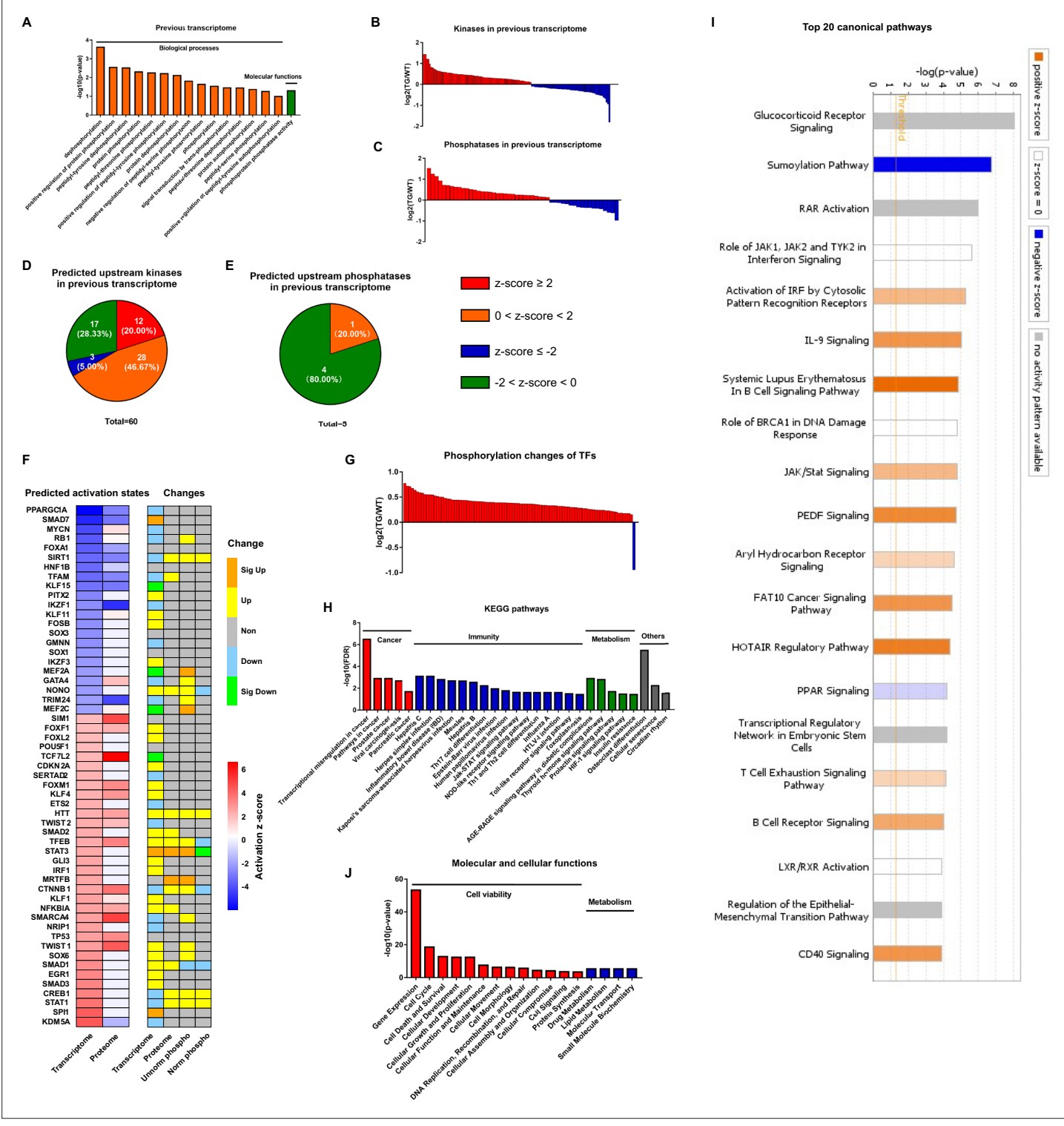

**Figure 4.** Coordination of phosphoproteome and transcriptome. (**A**) Gene ontology (GO) term enrichments of biological processes and molecular functions about phosphorylation and dephosphorylation from transcriptome. (**B–C**) Changes of kinases (**B**) and phosphatases (**C**) whose q-values are less than 0.05 in transcriptome. (**D–E**) States of upstream kinases (**D**) and phosphatases (**E**) are predicted based on transcriptome. Z-scores were calculated in ingenuity pathway analysis (IPA). Positive and negative z-scores indicate activated and inhibited states, respectively. The larger absolute values of z-score indicate more activated or inhibited states. (**F**) Heatmap of predicted upstream transcription regulators (left) and their changes in different omics datasets (right). Left: Fill color represents activation z-scores in transcriptome and total proteome. Red and blue indicate activated and inhibited states, respectively. Molecules are ranked by z-scores in transcriptome. Right: Fill color represents changes in transcriptome, total proteome, unnormalized

*Figure 4 continued on next page*

Figure 4 continued

phosphoproteome, and normalized phosphoproteome: orange: (p-value or q-value <0.05) and log2FC >0; yellow: (p-value or q-value ≥0.05) and log2FC >0; green: (p-value or q-value <0.05) and log2FC <0; light blue: (p-value or q-value ≥0.05); and log2FC <0; gray: unidentified in one of the omics. (**G**) Changes of differentially regulated transcription factors (TFs) with p-value <0.05 in the unnormalized phosphoproteome. (**H**) Kyoto Encyclopedia of Genes and Genomes (KEGG) pathway enrichments of the differentially regulated TFs. They are in four primary categories: cancer, immunity, metabolism, and others. (**I**) Top 20 canonical pathways enriched from the phosphoproteome using IPA. The darker the color is, the greater the absolute(z-score) is, and the more significantly enriched the pathway is. (**J**) Enriched molecular and cellular functions from the phosphoproteome using IPA. They are in two primary categories: cell viability and metabolism.

## Coordinated changes in protein expression and phosphorylation of the remodeled proteome and phosphoproteome of TGAC8

Having discovered the coordinated changes in the remodeled TGAC8 transcriptome and phosphoproteome (*Figure 4A–E*), we next sought to discover the extent to which changes in the remodeled phosphoproteome of the TGAC8 LV were coordinated with changes in its remodeled proteome. Numerous biological processes and molecular functions that were related to phosphorylation and dephosphorylation were enriched in the remodeled TGAC8 LV proteome (*Figure 5A*), as were numerous kinases and phosphatases (*Figure 5B–C* and File 10). Twenty-nine kinases that differed by genotype were predicted to be upstream regulators of the proteome, and most of these were predicted to be activated in TGAC8 vs. WT. Fewer phosphatases than kinases that differed by genotype were identified as upstream regulators of the remodeled TGAC8 LV proteome (*Figure 5D–E* and File 11).

One strategy adopted to probe deeper into the coordinated changes in the proteome and phosphoproteome of the TGAC8 LV was to normalize the phosphoproteome to the total proteome (in other words, to divide the phosphorylation expression by protein expression) (File 1: sheet4). Of the 3927 phosphoproteins identified in our mouse LV samples, 2164 were identified in both the proteome and phosphoproteome, and these were focused for subsequent analyses. After normalization, WT and TGAC8 samples manifested clear genotypic differences in PCA analysis (*Figure 5G*), and the phosphorylation status of numerous proteins normalized to the protein expression level still differed by genotype without cutoff (*Figure 5H*) and with cutoff (-log(p-value)>1.3 and absolute(log2(TG/WT))≥0.4) (*Figure 5I*). Even after normalizing the phosphorylation level to the protein expression level, the enrichments of canonical pathways in TGAC8 were quite consistent with those of unnormalized phosphoproteome (*Figure 5J* and File 6). Cardiac hypertrophy signaling (enhanced) and cardiac hypertrophy signaling took the top two places, emphasizing the important role of phosphorylation in the regulation of cardiac functions in response to chronic stress when coordinated with changes in the total proteome.

Many important stress-response signaling pathways were enriched from phosphoproteome, e.g., PKA signaling, ERK/MAPK signaling, insulin receptor signaling, and PI3K/AKT signaling (*Figure 6A*). Three of these pathways, PI3K/AKT, ERK/MAPK, and ubiquitin labeling, were also markedly enriched in our previous analysis of total proteome in TGAC8 LV (*Tarasov, 2022*). By integrating and comparing the enrichments from transcriptome, proteome, and phosphoproteome, we were able to identify the three signaling pathways in the TGAC8 LV, PI3K/AKT, ERK/MAPK, and ubiquitin labeling, that was regulated concordantly in transcription, translation, and PTM (phosphorylation) (*Figure 6B*). WB validated the activation of the PI3K/AKT signaling pathway in TGAC8, reproducing the results of our prior study using different antibodies (*Figure 6C–D* and *Figure 6—figure supplement 1A* (more significant)). We also demonstrated in that study that ERK/MAPK signaling was also activated in TGAC8 vs. WT (*Figure 6—figure supplement 1B–C*). Activation of PI3K/AKT and ERK/MAPK signaling is crucial for enhanced protein quality control (the increase in protein synthesis and degradation) in the TGAC8 LV (*Tarasov, 2022*; *Figure 6—source data 1D- G*). Although protein ubiquitination, an important process in protein degradation, was enriched in TGAC8 LV within IPA analysis, its activity could not be predicted on the basis of the present knowledge base of IPA. Its WB analysis, however, indicated that ubiquitination is indeed increased in TGAC8 vs. WT (*Figure 6E*).

We next sought to identify which molecules linked to PI3K/AKT signaling (one of the most important pathways in this mouse model) differed in TGAC8 vs. WT. Among the significantly changed upstream molecules, the majority were upregulated. AKT, a central molecule in the pathway, was also increased in proteome and phosphoproteome (*Figure 6F*). By searching and mapping the substrates of AKT,

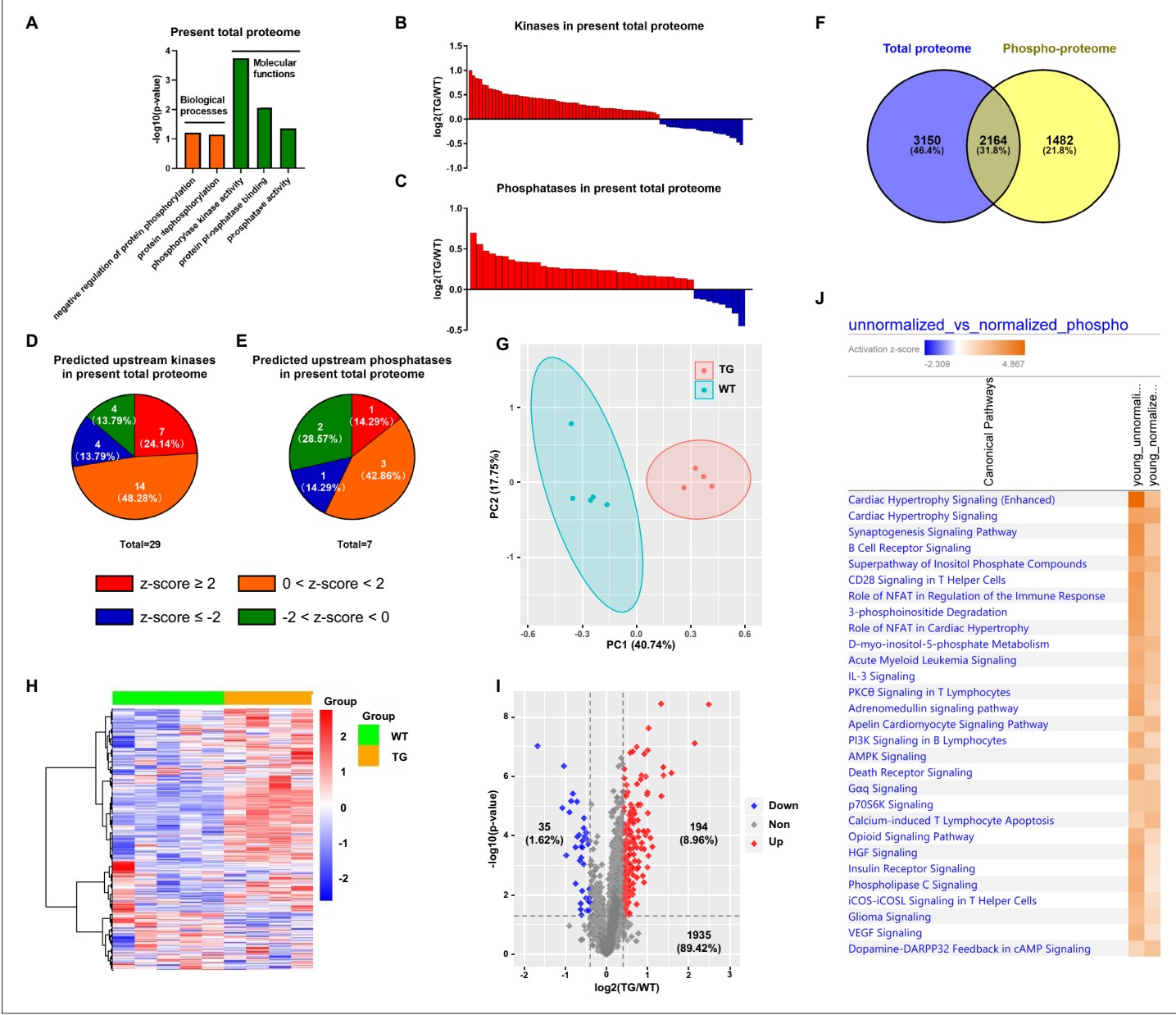

**Figure 5.** Analysis of normalized phosphoproteome. (**A**) Gene ontology (GO) term enrichments of biological processes and molecular functions about phosphorylation and dephosphorylation from the present proteome. (**B–C**) Changes of kinases (**B**) and phosphatases (**C**) whose q-values are less than 0.05 in previous proteome. (**D–E**) States of upstream kinases (**D**) and phosphatases (**E**) are predicted based on the present proteome. Z-scores were calculated in ingenuity pathway analysis (IPA). Positive and negative z-scores indicate activated and inhibited states, respectively. The larger absolute values of z-score indicate more activated or inhibited states. (**F**) Venn plot of new total proteome and unnormalized phosphoproteome that were conducted concurrently in the same platform. (**G**) Principal component analysis (PCA) plot of phosphoproteome normalized to total proteome. (**H**) Heatmap of expressions of phosphoproteins normalized to total proteins.(**I**) Volcano plot of normalized phosphoproteome. Red dots represent upregulated phosphoproteins with -log10(p-value)>1.3 and log2(fold change)≥0.4; blue dots represent downregulated phosphoproteins with -log10(p-value)>1.3 and log2(fold change)≤–0.4; gray dots represent nonsignificantly regulated phosphoproteins with -log10(p-value)≤1.3 or absolute(log2(fold change))<0.4. (**J**) Comparison of canonical pathway enrichments between unnormalized and normalized phosphoproteome. Orange fill colors indicate that all comparable pathways on the top are activated in both omics datasets. The darker the color is, the greater the absolute(z-score) is, and the more significantly enriched the pathway is.

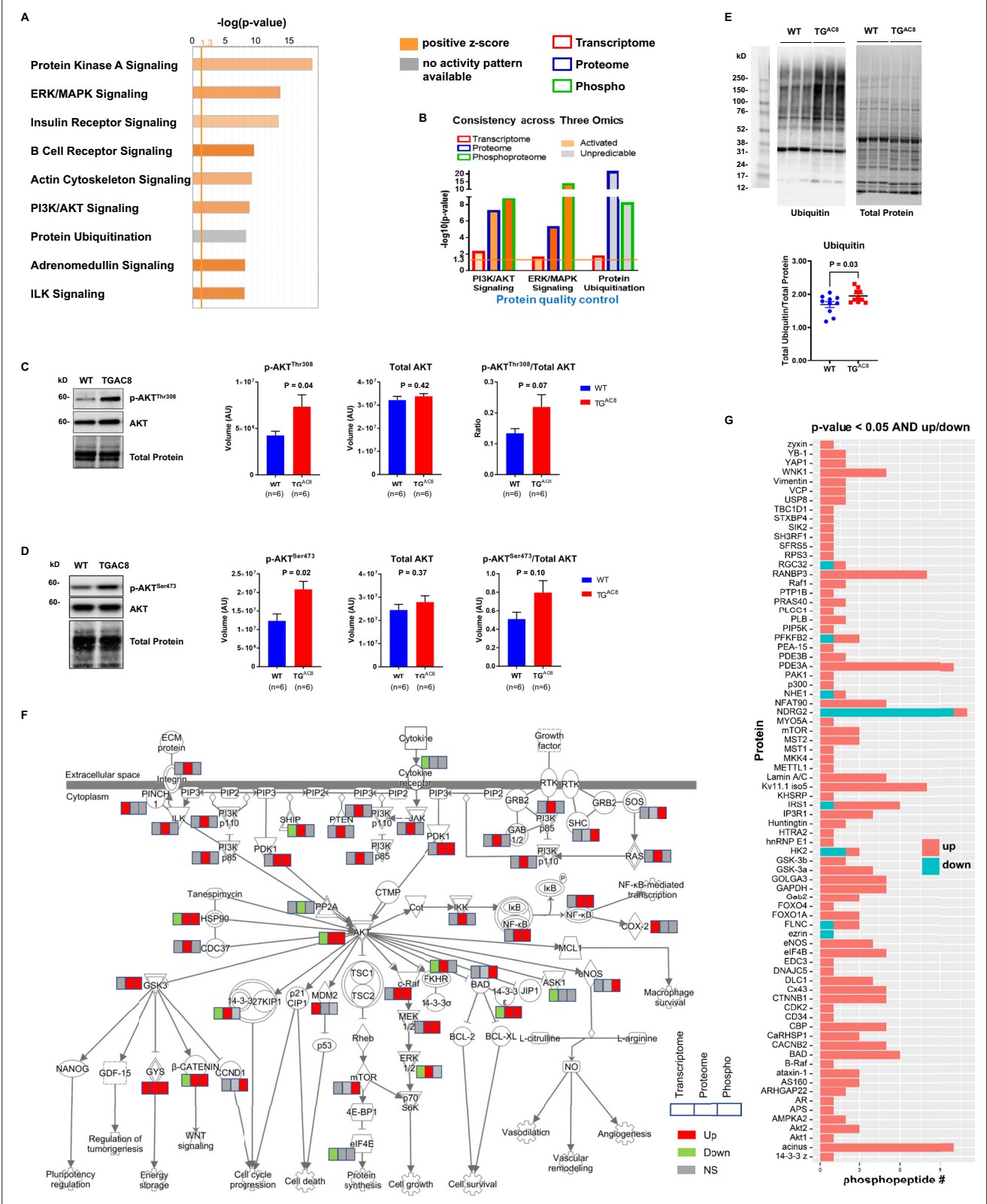

**Figure 6.** Coordinated regulation of transcriptome, proteome, and phosphoproteome on PI3K/AKT and ERK/MAPK signaling pathways and protein synthesis and degradation. (**A**) Canonical pathway enrichment of phosphoproteome only with cutoffp-value <0.05. The darker the color is, the greater the absolute (z-score) is, and the more significantly enriched the pathway is. (**B**) Comparison of PI3K/AKT signaling and protein ubiquitination pathways enriched from transcriptome (red outline), proteome (blue outline), and phosphoproteome (green outline). (**C–D**) Western blots of two phosphorylation

*Figure 6 continued on next page*

*Figure 6 continued*

sites on pan-AKT protein for verification of phosphorylation and activation of PI3K/AKT signaling pathway in cardiac-specific overexpression of adenylyl cyclase type 8 (TGAC8) vs. wild-type (WT). (**E**) Western blot of ubiquitination in TGAC8 vs. WT. (**F**) Pathway graph of PI3K/AKT signaling marked with changes in transcriptome (left box), proteome (middle box), and phosphoproteome (right box). Fill colors indicate changes: red and green indicate upregulation and downregulation, respectively; gray indicates nonsignificant regulation. (**G**) Phosphopeptide number of substrates of PI3K. Y-axis represents protein substates of PI3K; x-axis represents number of phosphopeptides. Red and blue bars indicate upregulated (p-value <0.05 and log2(fold change)>0) and downregulated (p-value <0.05 and log2(fold change)<0) phosphopeptides, respectively. Two-tailed unpaired Student's t-test with Welch's correction was conducted and p-values were shown in plots.

The online version of this article includes the following source data and figure supplement(s) for figure 6:

**Source data 1.** The full raw unedited gels and blots and figures with the uncropped gels or blots with the relevant bands associated with *Figure 6C and D*.

**Source data 2.** The full raw unedited gels and blots and figures with the uncropped gels or blots with the relevant bands associated with *Figure 6E*.

**Figure supplement 1.** Cross-verification of activated PI3K/AKT and ERK/MAPK signaling pathways and enhanced protein synthesis and degradation from our previous study.

we unraveled that the phosphorylation levels of AKT peptide substrates were mostly elevated in TGAC8 vs. WT (*Figure 6G*). Therefore, the downstream effects and functions of AKT signaling were activated in TGAC8 LV: (1) upregulating GSK3 to improve energy metabolism; (2) activating mTOR pathway to increase protein synthesis and cell growth; (3) promoting NF-κB mediated transcription; (4) upregulating BAD and eNOS only at phosphorylation level to enhance cell survival and cardiovascular functions. In summary, to adapt the heart to stress for improved survival, the TGAC8 reprograms the cardiac LV phosphoproteome to a large degree.

## Discordant regulation of the phosphoproteome and proteome in TGAC8 LV

Among the 443 proteins that differed by genotype in both protein phosphorylation and protein expression levels, 33 were changed discordantly at the two levels (in opposite directions, i.e. upregulated in proteome but downregulated in phosphoproteome, or vice versa) (*Figure 7A* and File 7: Sheet 1). We uploaded the 33 discordant proteins into STRING online software for functional analyses. The GO_CC (Cellular Component) analysis resulted in many **mitochondrial components**, indicating the involvement of phosphorylated proteins in mitochondria (*Figure 7B* and File 7: Sheet 2). Reactome pathway analysis showed that in addition to mitochondria, metabolic pathways were enriched significantly in TGAC8 (*Figure 7C* and File 7: Sheet 3). Mitochondria are the hub of metabolism and multiple metabolic processes are executed in mitochondria. Consistently, both metabolism and mitochondria-related processes were ranked the top processes in GO_BP (Biological Process) analysis (*Figure 7D* and File 7: Sheet 4). Therefore, the discordant part between total proteome and phosphoproteome indicated that the phosphorylation plays a critical role in the regulation of mitochondria and metabolism in the TGAC8, distinct from its role in regulation at the translational level.

## Deduced schematic of coordination of phosphoproteome with transcriptome and proteome in TGAC8 mouse LV

A deduced schematic integration of the remodeled phosphoproteome, proteome, and transcriptome of the TGAC8 LV is shown in *Figure 8*. The TGAC8 mouse initially induces a PKA-dependent phosphorylation via AC8-cAMP-PKA axis. Many cytoplasmic proteins including TFs are thus phosphorylated and then transported into the nucleus where TFs recognize and bind to the promoters to activate the transcription of more regulatory factors, including kinases, phosphatases, and TFs. The activation states of these factors are changed upon phosphorylated in cytosol, thereby activating additional kinases-controlled phosphorylation cascades and reprogramming the cardiac phosphoproteome comprehensively.

## Discussion

The major novel findings of the present study were differentially regulated phosphoproteins by genotype and the enrichment of numerous stress-response pathways within the reprogrammed TGAC8 LV,

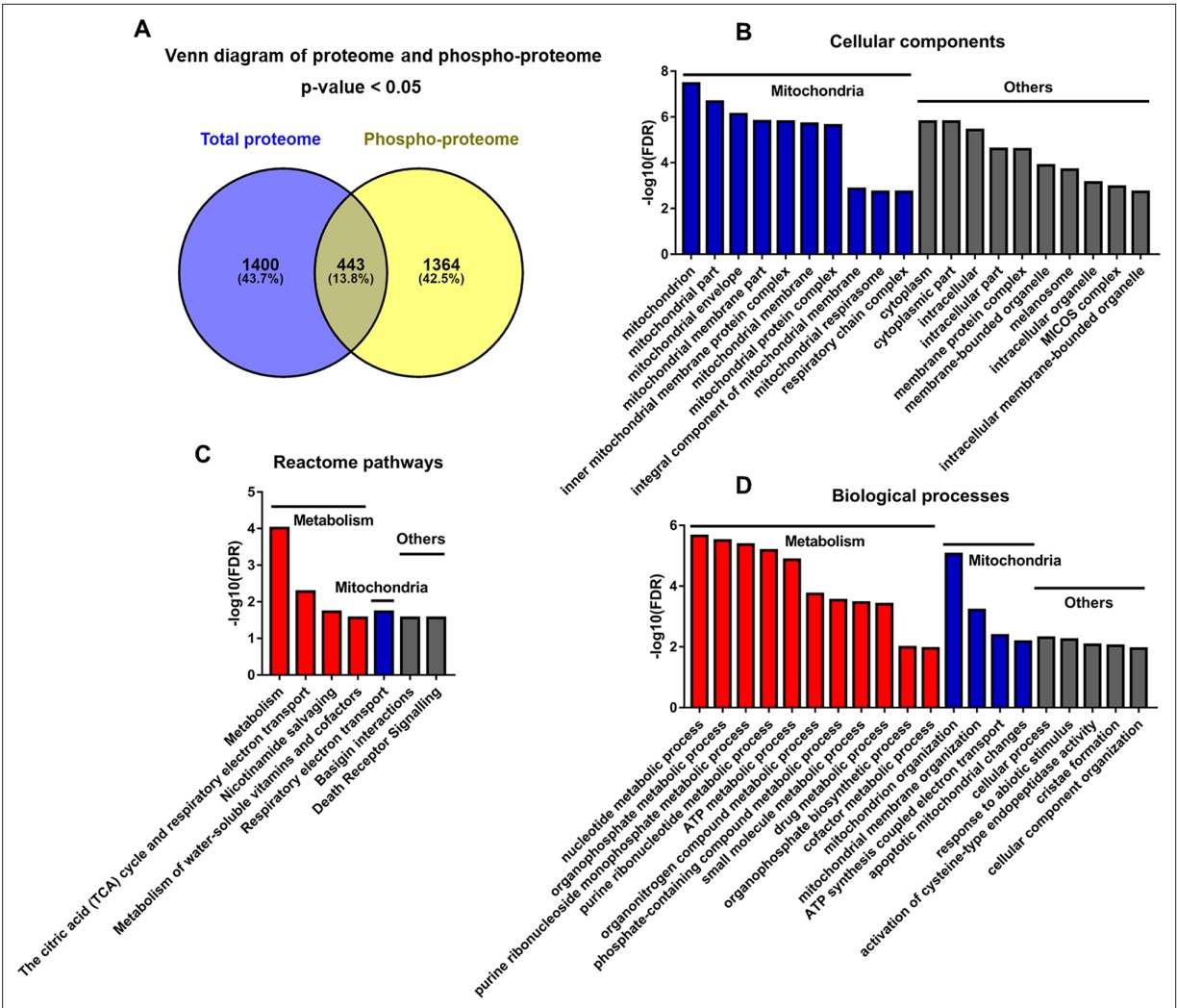

**Figure 7.** Analyses of proteins changed discordantly in proteome and phosphoproteome. (**A**) Venn diagram of significantly regulated proteins in total proteome and significantly regulated phosphoproteins in unnormalized phosphoproteome that were conducted concurrently in the same platform. (**B–D**) Functional analyses of proteins whose expressions are changed discordantly at total protein and phosphorylation levels. Mitochondria and others are enriched in cellular component analysis (**B**). Functions related to metabolism, mitochondria, and others are enriched in Reactome pathway (**C**) and biological process analyses (**D**).

including PKA, PI3K, and AMPK signaling pathways, their predicted upstream regulators (e.g. PDPK1, PAK1, and PTK2B), and downstream functions (e.g. cell viability, protein quality control, and metabolism). In addition to PKA, numerous other kinases and phosphatases were hyper-phosphorylated in TGAC8 vs. WT. Hyper-phosphorylated transcriptional factors in TGAC8 were associated with increased mRNA transcription, immune responses, and metabolic pathways. Integration of the phosphoproteome with its proteome and with the previously published TGAC8 transcriptome enabled the elucidation of cardiac performance and adaptive protection profiles that were coordinately regulated at PTM (phosphorylation), translational, and transcriptional levels. Many stress-response signaling pathways, i.e., PI3K/AKT, ERK/MAPK, and ubiquitin labeling, were consistently enriched and activated in the TGAC8 LV at transcriptional, translational, and PTM levels.

Most previous studies of PKA-dependent phosphorylation via the AC8-cAMP-PKA axis (typical substrate motif of PKA-dependent phosphorylation) focused on PKA phosphorylated targets as downstream effects of activated AC8. For example, AC8-induced cAMP activates PKA to increase the phosphorylation of many key proteins in the cardiovascular system, e.g., ryanodine-receptor-type 2 (RyR2), L-type $Ca^{2+}$ channels (LTCC), phospholamban (PLN or PLB), etc (*Tarasov, 2022*). Our study identified 29 phosphosites on RyR2, 15 phosphosites on LTCC alpha 1 and beta 2, and two phosphosites (S16

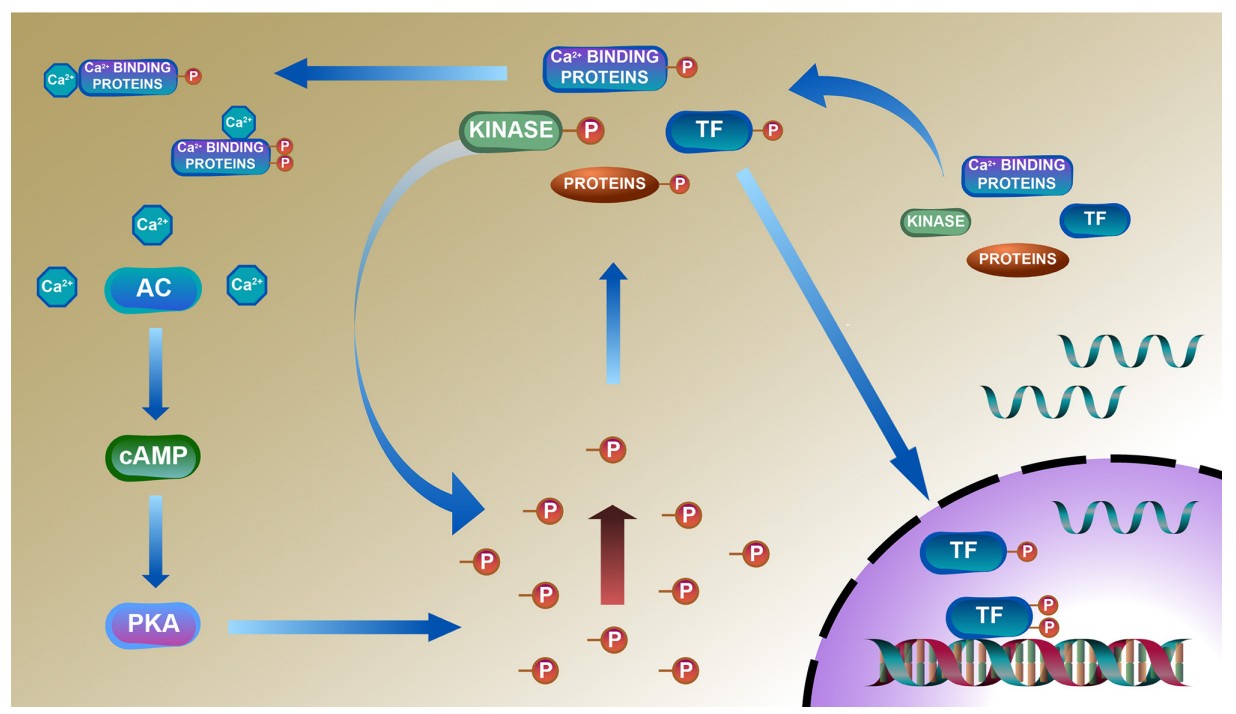

**Figure 8.** Deduced schematic of coordination of phosphoproteome with transcriptome and proteome in cardiac-specific overexpression of adenylyl cyclase type 8 (TGAC8) mouse left ventricle (LV). Based on our results, we deduced a mechanism of reprogramed phosphoproteome in TGAC8 mouse LV: First, TGAC8 mouse induces a PKA-dependent phosphorylation via AC8-cAMP-PKA axis. Many cytoplasmic proteins including transcription factors (TFs) are thus phosphorylated and then transported into the nuclear where TFs recognize and bind to the promotors to activate the transcription of more regulatory factors, including kinases, phosphatases, and TFs. The activation states of these factors are changed upon phosphorylated in the cytosol, thereby activating additional kinases-controlled phosphorylation cascades and reprogramming the cardiac phosphoproteome comprehensively. Left column: AC8-cAMP-PKA signaling; middle column: phosphorylation (phosphoproteome); lower right: transcription (transcriptome); upper right: translation (proteome).

and T17) on PLN, and, most interestingly, seven phosphosites on SERCA2 (Sarcoplasmic/endoplasmic reticulum calcium ATPase 2), with the majority of these increased in phosphorylation statuses in the TGAC8 LV (File 1: Sheet 1). It is worth noting that PLN is phosphorylated on S16 by PKA whereas on T17 by $Ca^{2+}$-calmodulin-dependent protein kinase (CaMKII) (*Wegener et al., 1989*).

Following phosphorylation, many cytoplasmic proteins, including TFs, are transported into the nucleus. TFs recognize and bind to the promoters to activate transcription, and many TFs were hyper-phosphorylated in TGAC8 vs.WT (*Figure 4F and G*). The phosphorylation of STAT1 (*Pilz et al., 2003*; *Wen et al., 1995*), STAT3 (*Wen et al., 1995*), HTT (*Kumar et al., 2014*), and CREB1 (*Wang et al., 2018*) is associated with the activation of their transcriptional activities. Elevated TFs phosphorylation and possibly activated transcription were consistent with the general increase in transcriptome in TGAC8 (*Tarasov, 2022*). The hyper-phosphorylated TFs might contribute to the improved immunity, metabolism, and cell viability in TGAC8 (*Figure 4H and I*). For example, STAT family proteins are transcription activators, and their phosphorylation is involved in inflammation and immune response (*Seif et al., 2017*; *Decker and Kovarik, 2000*). The present study identified increased phosphorylation levels in S727 on STAT1 and S726/S727 on STAT3 (File 1: Sheet 1), in line with our previous report which identified hyper-phosphorylation of components within inflammation and immune response signaling pathways such as JAK/STAT and JNK pathways (*Tarasov, 2022*).

Upon activation of RNA transcription, expression of numerous molecules, including TFs, kinases, and phosphatases was increased in TGAC8 vs WT, thereby, leading to activation of other kinase-controlled phosphorylation cascades in addition to PKA. For example, among the others, IPA predicted activation of PDPK1, PAK1, and PTK2B directed signaling in TGAC8 vs WT. In the context of the overexpression of AC8, these three kinases could be activated to phosphorylate multiple proteins to activate the PKA pathway (positive feedback) and the MAPK/ERK pathway (also known as Ras-Raf-MEK-ERK pathway)

to maintain heart health (*Figure 1E* and *Figure 1—figure supplement 2*). Previously, we showed WB results confirming MAPK/ERK pathway activation due to increased protein expression of p21 Ras and the elevated phosphorylation levels of c-Raf, MEK1/2, and ERK1/2 as well as their increased ratios of phosphorylation to protein expression (*Tarasov, 2022*). In the present study, S233 and S621/S641 on Isoform 2 of RAF proto-oncogene serine/threonine-protein kinase were hyper-phosphorylated in TGAC8 (File 1: Sheet 1). Importantly, the multifunctional molecule, GSK3B, was downstream of PDPK1 and PTK2B in the integrated regulation network. GSK3B is involved in metabolism and cardiac hypertrophy (*Beurel et al., 2015*; *Lal et al., 2015*). Interestingly, AC8 is required for glucose homeostasis and hypothalamic adaptation to a high-fat diet (*Raoux et al., 2015*). The phosphorylation of GSK3B is also a critical mediator in the convergence of cardioprotection signaling (*Juhaszova et al., 2004*; *Juhaszova et al., 2009*). We previously detected the increase in the protein expression of GSK3B (*Tarasov, 2022*) and here identified the hyper-phosphorylated S9 on GSK3B (File 1: Sheet 1).

Taken together, the TGAC8 upregulates phosphorylation to reprogram the heart. Based on the results in this study, combined with our recently published work on TGAC8 LV (*Tarasov, 2022*), we generated a comprehensive circuitry (*Figure 9*). Navigating this figure 'by the numbers:' marked, chronic, cardiac-specific overexpression of AC8 (1) chronically increases cAMP (2) and PKA-Ca$^{2+}$ signaling (3), resulting, not only in an incessantly increased cardiac performance, due to elevated heart rate and LV contractility (4), but also activation of growth factor signaling pathways (5), leading to increase in energy demand due to increased transcription and increased amino acid, nucleotide, protein, lipid and lipoprotein synthesis, increased protein quality control, including proteosome activity, autophagy, mitophagy, cytokine signaling and inflammation (6). Chronic high level of AC8 activation elicits numerous stress responses (8) including increased nutrient sensing with increased AMPK signaling (10 A), increased ROS scavenging, protection against cell death, and also induction of signaling to minimize the impact of AC-PKA signaling (10B). Shifts in energy production and utilization to deal with the marked energy demands of the TGAC8 heart (10, 12) include: enhancement of glucose metabolism is pronounced (11, 13), whilst utilization of fatty acids to produce energy is reduced (11) but an increase in glucose catabolism (10, 11) is accompanied by an increased anabolism (12), supported by increased aerobic glycolysis (11,13); increased utilization of the pentose phosphate shunt (5, 13) supports increased amino acid, nucleotide, protein and lipoprotein synthesis induced by growth factor signaling (5), resulting in an increase in LV biomass (14). This adaptive profile balances energy requirements necessary for an incessant marked increase in cardiac performance (4), for increased RNA transcription, protein translation, and quality control required for cardiac growth (14), and for other biological processes within the LV myocardium (6). This pattern of consilient adaptations (6-15) sustains an incessant increase in cardiac performance while conferring cardio-protection to the TGAC8 heart.

## Summary and future perspectives

The reprogrammed phosphoproteome activated stress response signaling pathways to defend against cardiac stress in young mice, improve cell viability, and suppress cell death. On one hand, the reprogramed phosphoproteome enhances transcription to improve the immunity and metabolism; on the other hand, it coordinates with the proteome to improve mitochondrial functions and metabolism. Finally, although we revealed the coordination of regulation at different levels, which provides a roadmap to guide the integration of transcription, translation, and PTM, our results set the stage for additional studies in this remarkable TGAC8 model: What kinases are specifically involved in the PTM identified here? How are these phosphorylated proteins specifically distributed in subcellular locations? Also, the specific details of how PDPK1, PAK1, and PTK2B directed signaling is activated by overexpression of AC8 requires further study. The bioinformatic analysis strategy here revealed a larger kinase-substrate network important in the remodeling of the TGAC8 heart. We plan to validate these kinase-substrate reactions in additional follow-up studies and provide the reliable mechanism in cardiovascular research.

The present study focuses on a comprehensive in silico analysis of functionally coherent and data-driven transcription, translation, and post-translation patterns that were well translated in the contribution to AC8-overexpression cardiac outcomes, while further investigation is warranted to see whether the hub pathway/function predictions via gene sets, phosphopeptides, and phosphoproteins could be used as biomarkers or impact the diagnostic specificity for transgenic AC8 cardiac health and

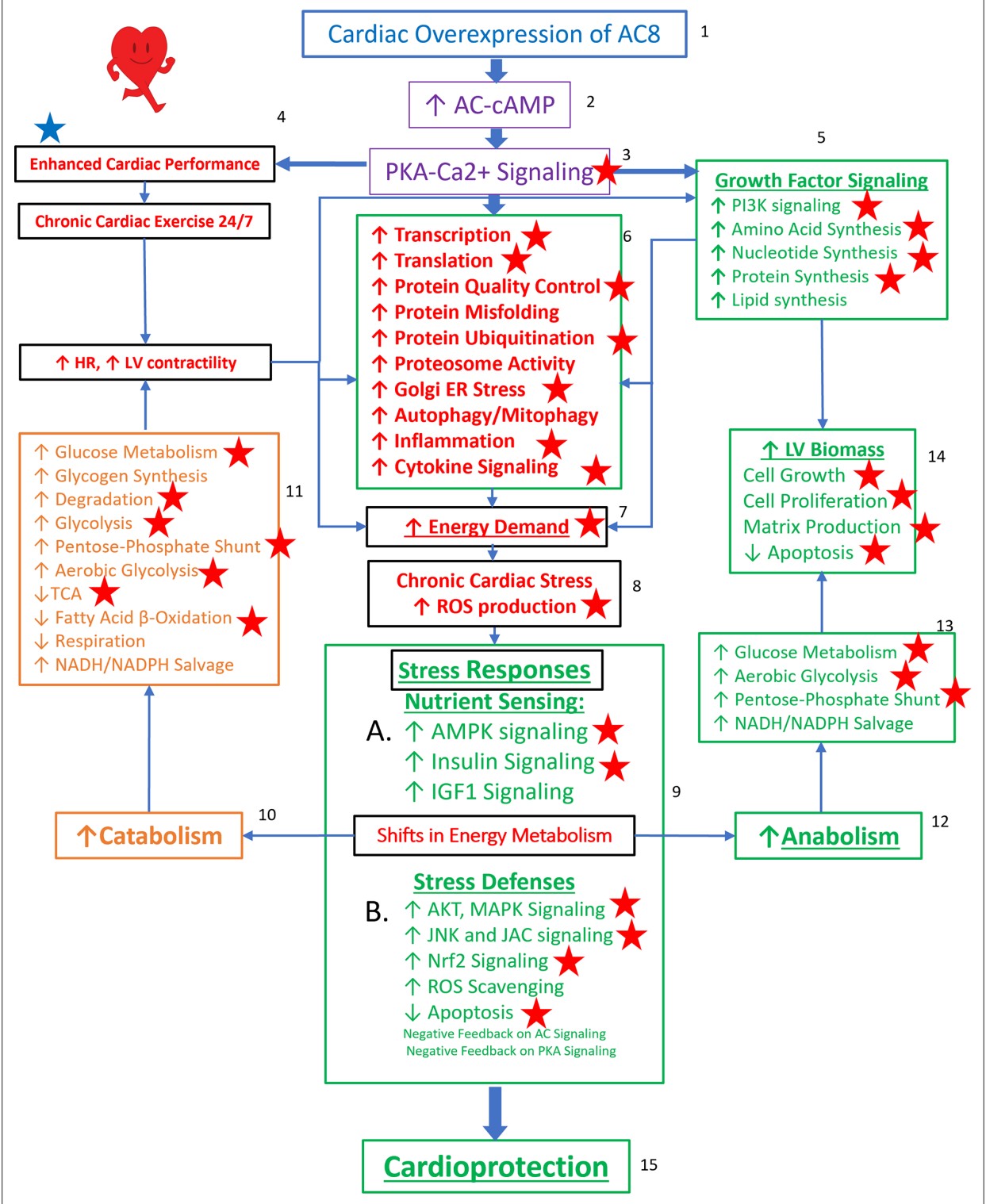

**Figure 9.** Reprogrammed concentric performance and protection circuitry in response to the constitutive challenge of marked cardiac-specific overexpression of adenylyl cyclase type 8 (TGAC8). Based on the present phosphoproteome, we optimized our previously deduced mechanism graph and labeled findings from the phosphoproteome using red stars. TGAC8 mouse induces a chronic cardiac exercise on the heart, manifested as increased heart rate and contractility accompanied with increased energy demand and cardiac biomass. Multiple signaling pathways, related to nutrient sensing, energy production, cardiac cellular stress, cardiac stress response, growth factor signaling, and cell proliferation, are potential downstream mediating the effect of the activated AC8-cAMP-PKA-Ca$^{2+}$ signaling in mouse left ventricle (LV). The altered cardiac functions were also observed associated with protein phosphorylation in our previous study and labeled in blue star here.

pathology, and even be used as therapeutic targets to enhance heart adaptivity, maintaining heart performance while avoiding cardiac dysfunction.

## Acknowledgements

We thank all members in the Laboratory of Cardiovascular Science, the National Institute on Aging, the National Institutes of Health, for technical assistance and discussion. This work utilized the computational resources of the NIH HPC Biowulf cluster (http://hpc.nih.gov).

## Additional information

### Funding

| Funder | Grant reference number | Author |
|---|---|---|
| National Institutes of Health | | Edward G Lakatta |

The funders had no role in study design, data collection and interpretation, or the decision to submit the work for publication.

### Author contributions

Jia-Hua Qu, Conceptualization, Data curation, Formal analysis, Investigation, Visualization, Methodology, Writing – original draft, Writing – review and editing; Khalid Chakir, Kirill V Tarasov, Allwin Jennifa Silvester, Writing – review and editing; Daniel R Riordon, Maria Grazia Perino, Validation, Writing – review and editing; Edward G Lakatta, Conceptualization, Resources, Funding acquisition, Writing – original draft, Project administration, Writing – review and editing

### Author ORCIDs

Jia-Hua Qu http://orcid.org/0000-0003-3912-656X
Kirill V Tarasov http://orcid.org/0000-0001-7799-4670
Edward G Lakatta https://orcid.org/0000-0002-4772-0035

### Ethics

All studies were performed in accordance with the Guide for the Care and Use of Laboratory Animals published by the National Institutes of Health (NIH Publication no. 85-23, revised 1996). The experimental protocols were approved by the Animal Care and Use Committee of the National Institutes of Health (protocol #441-LCS-2016).

Reviewer #1 (Public Review): https://doi.org/10.7554/eLife.88732.3.sa1
Reviewer #2 (Public Review): https://doi.org/10.7554/eLife.88732.3.sa2
Author Response https://doi.org/10.7554/eLife.88732.3.sa3

## Additional files

### Supplementary files
• MDAR checklist

### Data availability

The raw data and the quantification statistics of the protein peptides and phosphopeptides have been deposited in the MassIVE database (ID: MSV000089850).

The following dataset was generated:

| Author(s) | Year | Dataset title | Dataset URL | Database and Identifier |
|---|---|---|---|---|
| Qu JH, Chakir K, Tarasov KV, Lakatta EG | 2024 | The left ventricular phosphoproteome in mouse with cardiac-specific adenylyl cyclase type 8 overexpression | https://massive.ucsd.edu/ProteoSAFe/dataset.jsp?task=fcde724789094a0cb3f28b34070546d2 | MassIVE, MSV000089850 |

The following previously published dataset was used:

| Author(s) | Year | Dataset title | Dataset URL | Database and Identifier |
|---|---|---|---|---|
| Tarasov KV, Chakir K, Tarasova YS, Lakatta EG | 2022 | RNA-Seq analyses of TGAC8 and WT mouse left ventricles | https://www.ncbi.nlm.nih.gov/geo/query/acc.cgi?acc=GSE205234 | NCBI Gene Expression Omnibus, GSE205234 |

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
