## [Editor Report · eLife assessment]

This study describes **important** results from cardiac-specific overexpression of adenylyl cyclase type 8 (TGAC8) mice that was integrated with transcriptomic and proteomic evidence. The paper **convincingly** provides new insights into how one can interpret signals from visceral organs.

---

## [Referee Report · Reviewer #1 (Public Review)]

In this study the authors attempt to describe alterations in gene expression, protein expression, and protein phosphorylation as a consequence of chronic adenylyl cyclase 8 overexpression in a mouse model. This model is claimed to have resilience to cardiac stress.

Major strengths of the study include (1) the large dataset generated which will have utility further scientific inquiry for the authors and others in the field, (2) the innovative approach of using cross-analyses linking transcriptomic data to proteomic and phosphoproteomic data. One weakness is the lack of a focused question and clear relevance to human disease. These are all critical biological pathways that the authors are studying and essentially, they have compiled a database that could be surveyed to generate and test future hypotheses.

---

## [Referee Report · Reviewer #2 (Public Review)]

In this study, the investigators describe an unbiased phosphoproteomic analysis of cardiac-specific overexpression of adenylyl cyclase type 8 (TGAC8) mice that was then integrated with transcriptomic and proteomic data. The phosphoproteomic analysis was performed using tandem mass tag-labeling mass spectrometry of left ventricular (LV) tissue in TGAC8 and wild-type mice. The initial principal component analysis showed differences between the TGAC8 and WT groups. The integrated analysis demonstrated that many stress-response, immune, and metabolic signaling pathways were activated at transcriptional, translational, and/or post-translational levels.

The authors are to be commended for a well-conducted study with quality control steps described for the various analyses. The rationale for following up on prior transcriptomic and proteomic analyses is described. The analysis appears thorough and well-integrated with the group's prior work. Confirmational data using Western blot is provided to support their conclusions. Their findings have the potential of identifying novel pathways involved in cardiac performance and cardioprotection.

---

## [Author Response]

The following is the authors’ response to the original reviews.

**Reviewer #1 (Public Review):**
In this study, the authors attempt to describe alterations in gene expression, protein expression, and protein phosphorylation as a consequence of chronic adenylyl cyclase 8 overexpression in a mouse model. This model is claimed to have resilience to cardiac stress.Major strengths of the study include (1) the large dataset generated which will have utility for further scientific inquiry for the authors and others in the field, (2) the innovative approach of using cross-analyses linking transcriptomic data to proteomic and phosphoproteomic data. One weakness is the lack of a focused question and clear relevance to human disease. These are all critical biological pathways that the authors are studying and essentially, they have compiled a database that could be surveyed to generate and test future hypotheses.

Thank you for your efforts to review our manuscript, we are delighted to learn that you found our approach to link transcriptomic, proteomic and phosphoproteome data in our analysis to be innovative. Your comment that we have not focused on a question with clear relevance to human disease is “right on point!”

During chronic pathophysiologic states e.g., chronic heart failure (CHF) in humans, AC/cAMP/PKA/Ca2+ signaling increases progressively the degree of heart failure progresses, leading to cardiac inflammation, mediated in part, by cyclic-AMP- induced up- regulation of renin-angiotensin system (RAS) signaling. Standard therapies for CHF include β-adrenoreceptor blockers and RAS inhibitors, which although effective, are suboptimal in amelioration of heart failure progression. One strategy to devise novel and better therapies for heart failure, would be to uncover the full spectrum of concentric cardio- protective adaptations that becomes activated in response to severe, chronic AC/cAMP/PKA/Ca2+ -induced cardiac stress.

We employed unbiased omics analyses, in our prior study (https://elifesciences.org/articles/80949v1) of the mouse harboring cardiac specific overexpression of adenylyl cyclase type 8 (TGAC8), and identified more than 2,000 transcripts and proteins, comprising a broad array of biological processes across multiple cellular compartments, that differed in TGAC8 left ventricle compared to WT. These bioinformatic analyses revealed that marked overexpression of AC8 engages complex, concentric adaptation "circuity" that has evolved in mammalian cells to confer resilience to stressors that threaten health or life. The main human disease category identified in these analyses was Organismal Injury and Abnormalities, suggesting that defenses against stress were activated as would be expected, in response to cardiac stress. Specific concentric signaling pathways that were enriched and activated within the TGAC8 protection circuitry included cell survival initiation, protection from apoptosis, proliferation, prevention of cardiac-myocyte hypertrophy, increased protein synthesis and quality control, increased inflammatory and immune responses, facilitation of tissue damage repair and regeneration and increased aerobic energetics. These TGAC8 stress response circuits resemble many adaptive mechanisms that occur in response to the stress of disease states and may be of biological significance to allow for proper healing in disease states such as myocardial infarction or failure of the heart. The main human cardiac diseases identified in bioinformatic analyses were multiple types cardiomyopathies, again suggesting that mechanisms that confer resilience to the stress of chronic increased AC-PKA-Ca2+ signaling are activated in the absence of heart failure in the super-performing TGAC8 heart at 3-months of age.

In the present study, we performed a comprehensive in silico analysis of transcription, translation, and post-translational patterns, seeking to discover whether the coordinated transcriptome and proteome regulation of the adaptive protective circuitry within the AC8 heart that is common to many types of cardiac disease states identified in our previous study (https://elifesciences.org/articles/80949v1) extends to the phosphoproteome.

**Reviewer #2 (Public Review):**
In this study, the investigators describe an unbiased phosphoproteomic analysis of cardiac-specific overexpression of adenylyl cyclase type 8 (TGAC8) mice that was then integrated with transcriptomic and proteomic data. The phosphoproteomic analysis was performed using tandem mass tag-labeling mass spectrometry of left ventricular (LV) tissue in TGAC8 and wild-type mice. The initial principal component analysis showed differences between the TGAC8 and WT groups. The integrated analysis demonstrated that many stress-response, immune, and metabolic signaling pathways were activated at transcriptional, translational, and/or post-translational levels.The authors are to be commended for a well-conducted study with quality control steps described for the various analyses. The rationale for following up on prior transcriptomic and proteomic analyses is described. The analysis appears thorough and well-integrated with the group's prior work. Confirmational data using Western blot is provided to support their conclusions. Their findings have the potential of identifying novel pathways involved in cardiac performance and cardioprotection.

Thank you for your efforts to review our manuscript, we are delighted to learn that you found our approach to link transcriptomic, proteomic and phosphoproteome data in our analysis. We are delighted that you found our work to be well-conducted, to have been well performed, and that our analysis was thorough and well-integrated with our prior work in this arena and that are findings have the potential of identifying novel pathways involved in cardiac performance and cardioprotection.

**Reviewer #1 (Recommendations For The Authors):**
I humbly suggest that the authors reconsider the title, as it could be more clear as to what they are studying. Are the authors trying to highlight pathways related to cardiac resilience? Resilience might be a clearer word than "performance and protection circuitry".

Thank you for this important comment. We have revised the title accordingly: Reprogramming of cardiac phosphoproteome, proteome and transcriptome confers resilience to chronic adenylyl cyclase-driven stress.

Perhaps the text can be reviewed in detail by a copy-editor, as there are many grammatically 'awkward' elements (for example, line 56: "mammalians" instead of mammals), inappropriate colloquialisms (for example, line 73: "port-of-call"), and stylistic unevenness that make it difficult to read.

We have reviewed the text in detail, with the assistance of a copy editor, in order to identify and correct awkward elements and to search for other colloquialisms. Finally, although “stylistic unevenness” to which you refer may be difficult for us to identify during our re-edits, we have tried our best to identify and revise them.

The best-written sections are the first few paragraphs of the discussion section, which finally clarify why the TGAC8 mouse is important in understanding cardiac resilience to stress and how the present study leverages this model to disentangle the biological processes underlying the resilience. I wish this had been presented in this manner earlier in the paper, (in the abstract and introduction) so I could have had a clearer context in which to interpret the data. It would also be helpful to point out whether the TGAC8 mouse has any correlates with human disease.

Thank you for this very important comment. Well put! In addition to recasting the title to include the concept of resilience, we have revised both the abstract and introduction to feature what you consider to be important to the understanding of cardiac resilience to stress, and how the present study leverages this model to disentangle the biological processes underlying the resilience.

**Reviewer #2 (Recommendations For The Authors):**
1. How were the cutoffs determined to distinguish between upregulated/downregulated phosphoproteins and phosphopeptides?

Thank you for this important question. We used the same criteria to distinguish differences between TGAC8 and WT for unnormalized and normalized phosphoproteins, -log10(p-value) > 1.3, and log2FoldChange <= -0.4 (down) or log2FoldChange >= 0.4 (up), as stated in the methods section, main text and figure legend. The results were consistent across all analyses and selectively verified by experiments.

1. Were other models assessed for correlation between transcriptome and phosphoproteome other than a linear relationship of log2 fold change?

Thank you for this comment. In addition to a linear relationship of log2 fold change of molecule expression, we also compared protein activities, e.g., Fig 4F, and pathways enriched from different omics, e.g., Fig 3D, 5J, 6B and 6F.

1. Figures 1A and 5G seem to show outliers. How many biological and technical replicates would be needed to minimize error?

Thank you for the question. Figures 1A and 5G were PCA plots which, as expected, manifested some genetic variability among the same genotypes. The PCA plots, however, are useful in determining how the identified items separated, both within and among genotypes. For bioinformatics analysis such as ours, 4-5 samples are sufficient to accomplish this, as demonstrated by separation, by genotype, of samples in PCA. Thus, in addition to discovery of true heterogeneity among the samples, our results are still able to robustly discover the true differences between the genotypes.

1. Were the up/downregulated genes more likely to be lowly expressed (which would lead to larger log2 changes identified)?

In response to your query, we calculated the average expression of phosphorylation levels across all samples to observe whether they were expressed in low abundance in all samples.We also generated the MA plots, an application of a Bland–Altman plot, to create a visual representation of omics data. The MA plots in Author response image 1 illustrate that the target molecules with significantly changed phosphorylation levels did not aggregate within the very low abundance. To confirm this conclusion, we adopted two sets of cutoffs: (1) change: -log10(p-value) > 1.3, and log2FoldChange < 0 (down) or log2FoldChange > 0 (up); and (2) change_2: -log10(p-value) > 1.3, and log2FoldChange <= -0.4 (down) or log2FoldChange >= 0.4 (up).

1. "We verified some results through wet lab experiments" in the abstract is vague.

Thank you for the good suggestion. What we meant to indicate here was that identified genotypic differences in selected proteins, phosphoproteins and RNAs discovered in omics were verified by western blots, protein synthesis detection, proteosome activity detection, and protein soluble and insoluble fractions detection. However, we have deleted the reference to the wet lab experiments in the revised manuscript.

1. There are minor syntactical errors throughout the text.

Thank you very much for the suggestion. As noted in our response, we have edited and revised those errors throughout the text.